# Apparent competition drives community-wide parasitism rates and changes in host abundance across ecosystem boundaries

Carol M. Frost[1,†], Guadalupe Peralta[1,†], Tatyana A. Rand[2], Raphael K. Didham[3,4], Arvind Varsani[1,5,6,7] & Jason M. Tylianakis[1,8]

Species have strong indirect effects on others, and predicting these effects is a central challenge in ecology. Prey species sharing an enemy (predator or parasitoid) can be linked by apparent competition, but it is unknown whether this process is strong enough to be a community-wide structuring mechanism that could be used to predict future states of diverse food webs. Whether species abundances are spatially coupled by enemy movement across different habitats is also untested. Here, using a field experiment, we show that predicted apparent competitive effects between species, mediated via shared parasitoids, can significantly explain future parasitism rates and herbivore abundances. These predictions are successful even across edges between natural and managed forests, following experimental reduction of herbivore densities by aerial spraying of insecticide over 20 hectares. This result shows that trophic indirect effects propagate across networks and habitats in important, predictable ways, with implications for landscape planning, invasion biology and biological control.

[1] Centre for Integrative Ecology, School of Biological Sciences, University of Canterbury, Private Bag 4800, Christchurch 8140, New Zealand. [2] USDA-ARS Northern Plains Agricultural Research Laboratory, Sidney, Montana 59270, USA. [3] School of Animal Biology, The University of Western Australia, 35 Stirling Highway, Crawley Western Australia 6009, Australia. [4] CSIRO Land & Water, Centre for Environment and Life Sciences, Underwood Ave, Floreat Western Australia 6014, Australia. [5] Biomolecular Interaction Centre, University of Canterbury, Private Bag 4800, Christchurch 8140, New Zealand. [6] Structural Biology Research Unit, Department of Clinical Laboratory Sciences, University of Cape Town, Observatory, Cape Town 7700, South Africa. [7] Department of Plant Pathology and Emerging Pathogens Institute, University of Florida, Gainesville, Florida 32611, USA. [8] Department of Life Sciences, Imperial College London, Silwood Park Campus, Buckhurst Road, Ascot, Berkshire SL5 7PY, UK. † Present addresses: Department of Forest Ecology and Management, Swedish University of Agricultural Sciences, Skogsmarksgränd, 90183 Umeå, Sweden (C.M.F.); Instituto Argentino de Investigaciones de las Zonas Áridas, CONICET, CC 507, 5500 Mendoza, Argentina (G.P.). Correspondence and requests for materials should be addressed to C.M.F. (email: carol7frost@gmail.com) or to J.M.T. (email: jason.tylianakis@canterbury.ac.nz).

Communities frequently experience population reductions (for example, via harvesting or native species decline) or species additions (during invasions, biological control or species range shifts). These changes can directly impact consumers or prey of the affected species[1]. However, population changes may also indirectly affect the entire community via longer pathways across the interaction network[2,3]. Unfortunately, these many subtle indirect interactions[4,5] limit our ability to predict the community-wide consequences of changes in abundance of a focal species, because they can significantly alter population growth and persistence[6,7], and even affect species abundances and distributions as strongly as direct feeding interactions[8]. In particular, the dynamics of different prey species can be linked via shared enemies (predators or parasitoids), even if the prey never compete directly for resources[9]. That is, an increase in the population of one prey species can cause a decrease in the population of other prey species by driving an increase in shared enemy abundance and attack rates or changes to enemy behaviour[6]. This phenomenon, termed 'apparent competition'[9], can be common in food webs[10], where it may be the most important indirect interaction affecting pairwise species dynamics[11].

There are several mechanisms by which enemy responses to prey population growth can cause apparent competition. The timescale over which these mechanisms occur can vary from within one prey generation, through aggregative[12] or functional responses[13], to between prey generations, through numerical responses[6], though distinguishing between them for an entire community under field conditions is unfeasible, and we do not attempt to do this here. Apparent mutualism is equally possible, if the shared enemy can be satiated or switch to the most abundant prey species in the short term[9], thereby releasing less-abundant prey from consumer pressure. Apparent mutualism could occur over the longer term if the population of one prey species cycles, such that it repeatedly satiates an enemy, and thus repeatedly alleviates consumer pressure on another prey species that shares the enemy[14]. However, fewer empirical examples of apparent mutualism have been documented[13,15,16]. Despite numerous isolated examples of apparent competition between species pairs[7,13,17], it remains untested whether the simultaneous pairwise effects of apparent competition across all species within a food web are strong enough to detectably affect population dynamics of all species, even amidst the network of direct interactions among them. The ability to predict the effects of changing population densities on interactions among all other species would be invaluable in addressing some of the most pressing questions in ecology and global environmental change, such as how ecosystems will respond to native species decline or species invasions.

Community-wide predictions of indirect interactions are further hindered by species movement among habitats. Global land-use change creates mosaic landscapes of managed and remnant natural habitats, and consumer movement among habitats is predicted to drive resident prey species dynamics through direct and indirect effects[18–20]. Yet, it remains untested whether mobile enemies dynamically couple herbivore assemblages in multiple habitats via apparent competition, as predicted by theory[18]. Entire suites of enemies can 'spill over' across habitat boundaries[21], and if they couple prey dynamics in the two habitats, this process could be an important mechanism by which anthropogenic habitats impact entire food webs in natural and managed areas throughout the landscape[20].

Here we test: (1) whether apparent competition influences community-wide parasitism rates and changes in herbivore abundance in host–parasitoid interaction networks (food webs) at the interface between native and plantation forests. In so doing,

we also test: (2) whether the future parasitism rate and abundance of each herbivore host species in the community can be predicted from quantitative food-web data on parasitoid overlap between hosts, as well as information about changes in abundance of all other hosts. We further test: (3) whether such predictions are possible across a habitat edge, or whether the edge hinders parasitoid movement or changes parasitoid–host selection such that predicted apparent competitive linkages between herbivore populations on either side of the edge are not realized. We conducted a simultaneous study of adult parasitoid movement between the two forest habitats considered here[21]. That study showed that parasitoids of many of the same species considered here moved between habitats throughout the season. However, more individuals moved from plantation to native forest than in the other direction, likely due to the higher productivity of plantation relative to native forest[21]. Thus, it could be that apparent competitive effects are asymmetrical between habitats, with stronger effects from herbivores in plantation forest on herbivores in native forest.

Our approach (see overview in Fig. 1) was to first determine a regional measure of shared parasitism (that is, the potential for apparent competition) between each pair of herbivore species (foliage-dwelling Lepidoptera larvae) in the system, by collecting quantitative food-web data (that is, numbers and identities of parasitoids attacking each host species) from a set of replicated training sites (Fig. 1a). Each site comprised samples from either side of a habitat edge between plantation *Pinus radiata* forest and native forest in New Zealand. To gather these data, we collected Lepidoptera larvae (caterpillars) and reared them to obtain and identify (morphologically and using DNA barcoding) the parasitoids (Hymenoptera, Diptera and Nematoda) that had attacked them. We carried out seven sampling rounds over two summers, to observe as many host–parasitoid interactions as possible (Fig. 1b,c). We predicted the potential for apparent competition between each pair of herbivore host species, using Müller *et al.*'s index[22] of the proportion of the parasitoids attacking one species that had recruited from the other species (Fig. 1d,e). This index quantifies the hypothesis that changes in abundance of each host species affect attack rates on every other host, proportionate to the number of parasitoid species they share and each host's contribution to the parasitoid pool. That is, for every pair of species that shares parasitoids, the species that produces more shared parasitoid individuals should have a larger apparent competitive impact on the other species. Thus, it incorporates the frequent asymmetry in potential for apparent competition between host species[17]. Muller *et al.*'s index[22], has been widely adopted to predict the potential for indirect interactions within a single location[23–26], but its predictive success has rarely been tested, and never across all the species in an assemblage. It has also never been tested across habitats. Nevertheless, two experimental studies have shown that it holds great promise for predicting indirect interactions among herbivores. First, Morris *et al.*[27] experimentally reduced the abundance of two leaf-miner species, and found reduced attack rates on other leaf miners with which they shared parasitoids. Similarly, Tack *et al.*[15] used a quantitative food web to predict interactions among three leaf-miner species, then experimentally increased the abundance of each species. They found cross-generation indirect interactions between some species as predicted, except that the effects were positive (that is, apparent mutualism[9]) rather than negative (apparent competition). Together these studies suggest that information on shared parasitoids can in principle be used to successfully predict indirect interactions between species. However, across entire food webs, many pathways of weak and strong, positive and negative indirect effects may render net outcomes unpredictable.

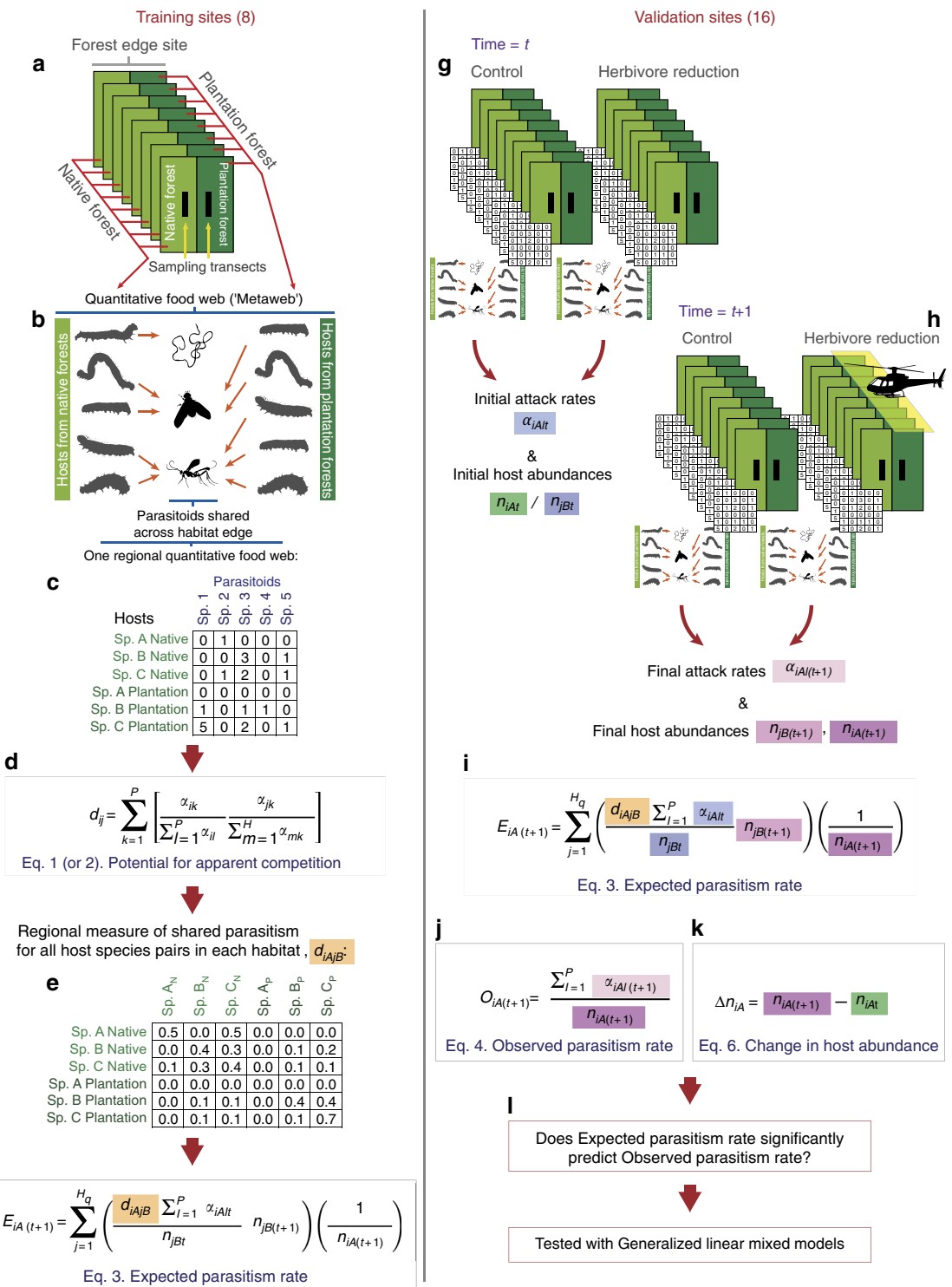

**Figure 1 | Summary of methods.** Sampling at training sites produced a measure of shared parasitism for each species pair among all species found in the region. (**a**) Lepidopteran larvae were collected along transects within native and plantation forest at eight forest edge sites. (**b**) Larvae were identified and reared to determine parasitism rates and parasitoid (Hymenoptera, Diptera and Nematoda) identities. (**c**) Data were pooled across sites and sampling dates, but kept separate by forest type, in order to produce one regional quantitative food web (host–parasitoid matrix), in which host habitat was explicit. (**d**) The potential for apparent competition, $d_{iAjB}$, was calculated for each host pair within and across habitats. (**e**) This $d_{iAjB}$ value was the regional measure of shared parasitism used to calculate Expected parasitism rate in equation (3) (**f,i**). (**g**) We sampled at control and herbivore reduction validation sites at two time steps (**g,h**), before and after the aerial spray herbivore reduction treatment, which occurred between time steps (**h**). This allowed measurement of the initial attack rates and initial host abundances (**g**) and changes in host abundances (**g,h**) necessary to calculate expected parasitism rate (**i**), as well as measurement of the final attack rates and final host abundances (**h**) necessary to calculate observed parasitism rates (**j**) and change in focal host abundances (**k**). We used generalized linear mixed models to test whether Expected parasitism rate significantly predicted observed parasitism rate and change in focal host abundance (**l**) at validation sites, and thus whether apparent competition structures host–parasitoid assemblages in a predictable manner.

Moreover, it remains unknown whether prey use by predators is consistent enough that food-web information generated in one location can be used to make accurate predictions about another, or whether entire prey communities in one habitat indirectly affect attack rates on, and therefore abundances of, those in adjacent habitats via mobile predators[21].

Thus, we tested whether our calculated predictions of apparent competition (Fig. 1e) could be used, along with data on changes in abundance of the potential apparent competitors, to predict parasitism rate and change in abundance of each herbivore species (each separately as a focal host) in the system. For this test, we selected another set of replicated validation sites (Fig. 1g). At these sites we again collected quantitative food-web data by sampling, rearing and identifying caterpillars and their parasitoids at two time steps: both before (Fig. 1g) and after a marked experimental herbivore reduction at half of the validation sites (Fig. 1h). This reduction was conducted by aerially spraying 2.5 hectares (ha) per site with a selective insecticide that targets Lepidoptera larvae (the hosts in our study). The experimental herbivore reduction allowed us to test whether our predictions of apparent competition performed equally for small, typical variation in host abundance (at control sites) as well as more dramatic changes, such as may occur during pest outbreaks or at plantation harvest. We then predicted expected parasitism rates (Methods equation (3); Fig. 1i) on all host species at the next time step (Fig. 1h), based on three pieces of information: (i) the regional measure of shared parasitism among hosts from training sites (Fig. 1d–f); (ii) initial attack rates on each host species at a given validation site (Fig. 1g); and (iii) the changes in abundance of all host species with which each focal host shares parasitoids (Fig. 1g,h). Finally, we used statistical models to test whether these expected parasitism rates predicted the observed parasitism rates at the time step after spraying (Fig. 1j), as well as changes in abundance between time steps (Fig. 1k). This test was conducted for each host species that occurred in the validation sites at the later time step (Fig. 1l).

We show that expected parasitism rate (our prediction of the final parasitism rate for each herbivore host species, based on the assumption of apparent competition with each other herbivore host species which occurred in the same site) significantly predicted both observed parasitism rate at the 'after' time step, and the change in abundance of a focal host between time steps. Expected parasitism rate predicted 31% of the variation in change in host abundance, and 15% of the variation in observed parasitism rate. Predictions worked equally well whether the focal host was in plantation or native forest, suggesting that in this system the habitat edge does not significantly hinder parasitoid movement or change host-selection behaviour, and suggesting that herbivore population changes in one forest type can have rapid and important effects on herbivore populations across the habitat edge.

## Results

**Data description.** Transect plus extra sampling at our training sites yielded 8321 caterpillars. Of these, 2725 individuals from 70 species in 13 families were successfully reared to moth or parasitoid emergence. These yielded 358 parasitism events by 46 species of Hymenoptera, Diptera and Nematoda parasitoids on 44 Lepidoptera species. These 358 parasitism events made up the data from which the training metaweb was constructed (Figs 1a–c and 2) and $d_{iAjB}$ was calculated (Methods equation (3), Fig. 1d–f). The metaweb had a binary connectance of 0.057, which is within the range of connectance values exhibited in published quantitative food webs[28].

Transect sampling at our validation sites yielded 5837 caterpillars that were identifiable to (morpho)species level, and

included 67 species. These made up the data from which we calculated all abundance terms ($n_{jB}$, $n_{iA}$) in equation (3) and equation (6) (Fig. 1g–k). Of these caterpillars, 2067 individuals from 60 species were successfully reared to moth or parasitoid emergence, yielding 263 parasitism events by 36 species of parasitoid, in Hymenoptera, Diptera and Nematoda on 25 species of Lepidoptera. Extra sampling yielded an additional 1121 identifiable caterpillars, of which 405 individuals from 65 species were successfully reared to moth or parasitoid emergence, yielding 40 additional parasitism events. The transect plus extra sampling total of 303 parasitism events by 37 species of parasitoids on 26 species of Lepidoptera made up the data from which $\alpha_{iAl(t)}$ was calculated in equations (3) and (5), from which $\alpha_{iAl(t+1)}$ and $n_{iA(t+1)}$ were calculated in equation (4), and from which $n_{iAt}$ was calculated in equation (5) (Fig. 1g–k).

**Regional potential for apparent competition among species.** From our regional quantitative food-web data from training sites, pooled over all sampling dates and sites (Fig. 2), we found that most parasitoid species were reared from hosts in both plantation and native forest (yellow interactions in Fig. 2), and few parasitoid species were specific to one forest type (green and purple interactions in Fig. 2). From these regional species interaction data, our calculations of potential for apparent competition between all host species pairs (Methods, equations (1) and (2), Figs 1d and 3) showed potential for apparent competition between 41 host species (circles connected by lines in Fig. 3). These calculations also showed potential for many cross-habitat apparent competitive interactions (grey lines in Fig. 3).

**Effects of experimental herbivore reduction.** Our experimental herbivore reduction at validation sites was successful in significantly reducing caterpillar numbers in treated plantation forests relative to control plantation forests (Fig. 4a; Supplementary Table 1). However, caterpillar numbers were also naturally lower in control plantation forests after the herbivore reduction treatment than before. The lower caterpillar numbers at control sites in the after time step was likely because of predation by invasive *Vespula* spp. wasps, which become very abundant in New Zealand plantation and native forests in the months during which the 'after' collection occurred[21], and which predate heavily on caterpillars[29]. Following the experimental reduction in herbivore abundance in the treated plantation forests, which occurred when many parasitoids were in their adult phase (Supplementary Fig. 1), it is possible that attack rates could have been briefly higher in native forest next to treated plantations, due to low host availability in plantations. However, we found no significant interaction effect (Fig. 4b; Supplementary Table 2), indicating that, overall, hosts in native forests were not attacked disproportionately more or less when their adjacent plantation was sprayed. This result is not surprising. We would not expect apparent competition effects to be strongly detectable in such an analysis, because it includes 'noise' in the attack rates such as unchanged parasitism rates for species in the native forest that do not share parasitoids across the habitat edge, or that share parasitoids with species whose abundances did not change drastically in response to the spray. Thus, to truly test the importance of host reduction, we needed to weight this test by host species changes in abundance combined with expectations of which host species might be expected to exert apparent competition on one another due to sharing of parasitoids.

**Apparent competition predicts community-wide parasitism.** Our higher resolution test, in which we calculated the expected

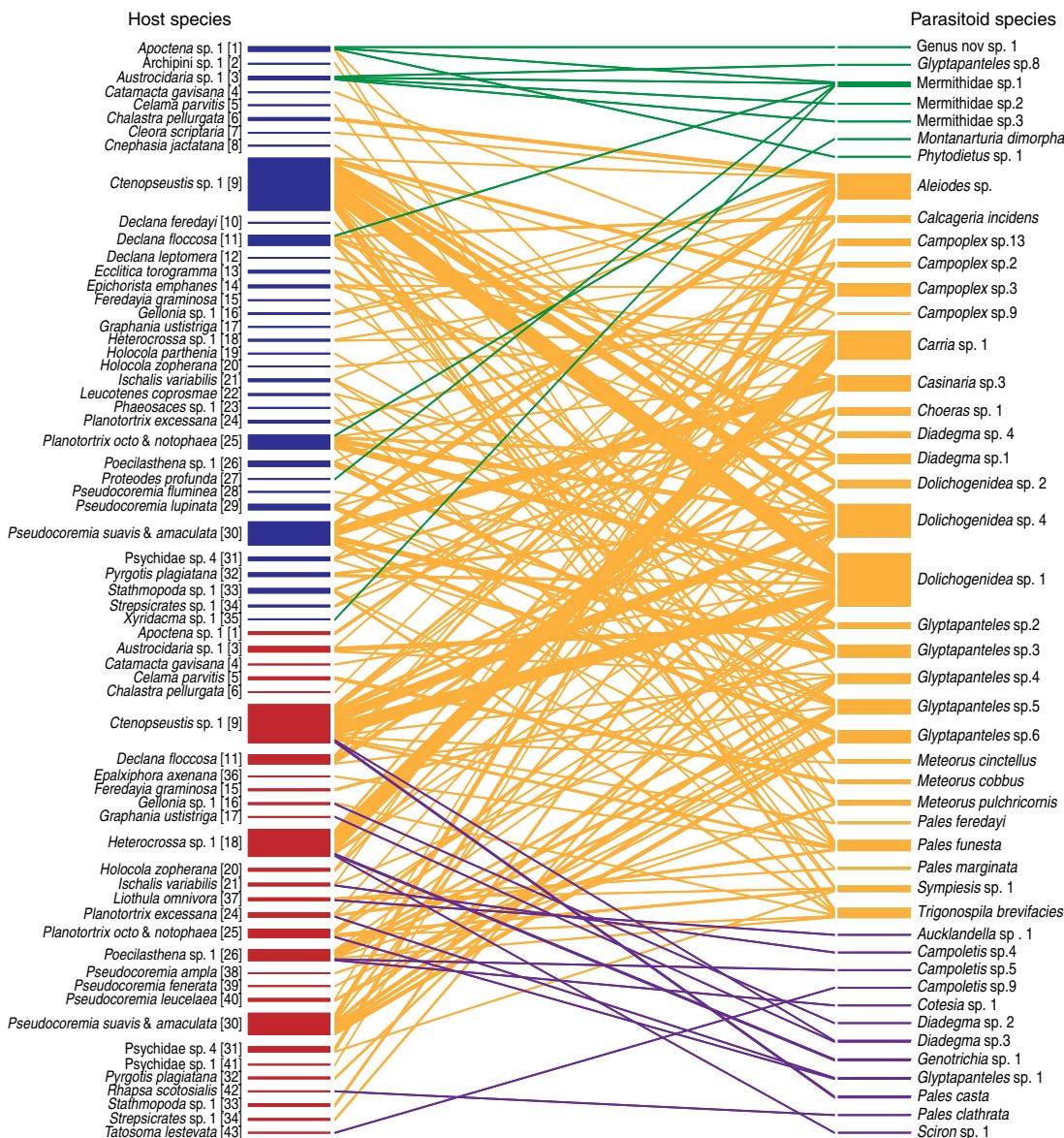

**Figure 2 | Regional metaweb built from quantitative food-web data.** Data were collected at habitat edges between native forest and exotic plantation forest and are pooled across eight training sites and seven sampling dates. Bars on the left represent herbivore host species in native forest (blue) or plantation forest (red), and bar thickness is proportional to number of parasitism events (358 total parasitism events from 2725 individual hosts reared). Bars on the right represent parasitoid species, and lines connecting parasitoids and host species denote parasitism, with parasitoids attacking hosts in native forest only (green), plantation forest only (purple) or both forest types (yellow). Parasitoid bar thickness is proportional to attack rate[56]. Numbers adjacent to host names correspond to the host numbers in Figure 2.

parasitism rates in the after time step for each host species (Methods equation (3)), showed that apparent competition significantly structured host–parasitoid assemblages within and across habitats in a predictable way. Our expected parasitism rate explained 15.6% of the variation in observed parasitism rates across host species ($z = 2.7$, $P = 0.007$), and the whole model (fixed plus random effects) explained 24.3% (Fig. 5a; Supplementary Table 3). This predictive ability was equally good for small natural or larger experimental changes in host abundance (the interaction between expected parasitism rate and herbivore reduction treatment was removed during model selection; that is, expected parasitism rate predicted observed parasitism rate equally well whether or not host abundance was drastically changed experimentally). Significant predictive ability remained even when within-habitat intraspecific effects were excluded from the calculation of expected parasitism rates

(Supplementary Table 4). Therefore the indirect effects at work could not simply be explained by within-habitat delayed density dependence. Importantly, this predictive capacity did not depend on the specific habitat of the focal host (the host habitat by expected parasitism rate interaction was removed during model selection; Fig. 5a, Supplementary Table 3). Rather, predicted apparent competitive effects were realized equally for hosts in either habitat, such that the habitat edge did not filter parasitoids and prevent some of the predicted interactions from occurring. Furthermore, this same predictive ability could not be achieved by using just initial parasitism rate for each focal host, calculated from quantitative food-web data from the initial time step at each site ($z = 1.11$, $P = 0.266$; Fig. 5b; Supplementary Table 5; Supplementary Fig. 2a,b provides equivalent figures to 5a,b with raw data). Rather, significant prediction required inclusion of the potential for apparent competition between each host species

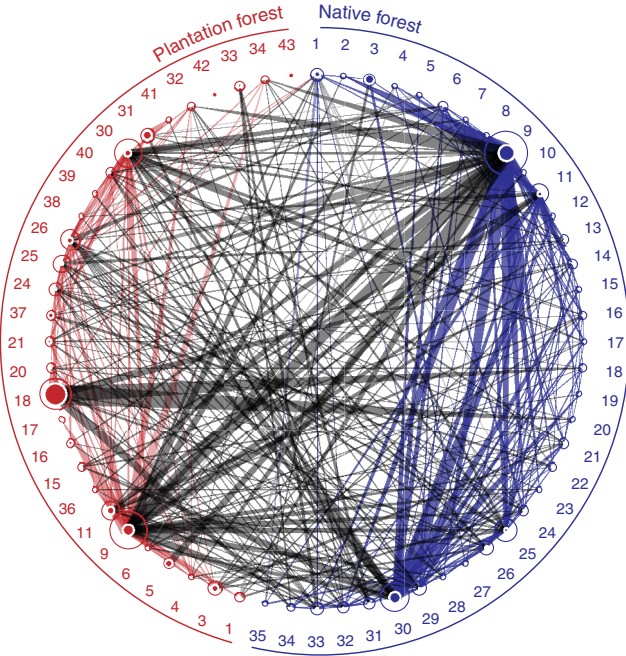

**Figure 3 | Parasitoid overlap graph for the region.** Data compiled from training sites (358 total parasitism events from 2725 individual hosts reared). This figure shows the potential for apparent competition between all parasitized herbivore host species in native forest (blue) and plantation forest (red). Numbers represent host species (see Fig. 2 for names), some of which are found in both habitats. Host circle size is proportional to the number of parasitoids recruiting from that host species, and circle fill represents the proportion of self-loops, that is, parasitoids attacking the same species from which they recruit. Lines between hosts represent sharing of parasitoids between those host species, with line thickness proportional to export of parasitoids[11,57]. Blue lines denote sharing of parasitoids by hosts within the native forest, red lines denote sharing of parasitoids by hosts within the plantation forest, and grey lines denote sharing of parasitoids between hosts in different habitats (the latter allowing cross-habitat apparent competition).

(the $d_{iAjB}$ in Methods equation (3)) and the changes in abundance of hosts with which the focal host shared parasitoids.

The number of species for which it was possible to calculate an expected parasitism rate at each site was far lower than the number of species collected at each site. In Fig. 5, each point represents a species within a site that was both collected and successfully reared at both time steps, and was parasitized in the first time step (see equation (3), Fig. 1i). However, the expected parasitism rates for these species within sites are based on the potential for apparent competition with every other host species that we collected in the site, whether or not it was parasitized or reared successfully, so the few data points are predicted by data from the entire network, and allowed us to test community-wide apparent competitive effects on future parasitism rates.

**Within- and cross-habitat predictions were successful**. In calculating the expected parasitism rate for each focal host, some of the predicted attacks were from parasitoids shared with hosts in the same habitat, and some were predicted to come from parasitoids shared with hosts in the adjacent habitat. We tested whether the cross-habitat component to the prediction was necessary to achieve predictive success, and also whether it was sufficient for predictive success. When we separated out the within-habitat versus cross-habitat contributions to expected parasitism rate for each host, and entered these as separate

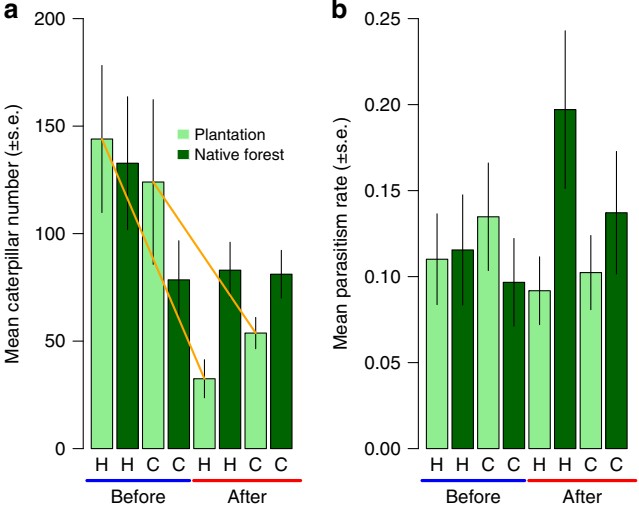

**Figure 4 | Community average effects of experimental herbivore reduction.** Experimental herbivore reduction (**a**) reduced caterpillar abundance in treated plantation forest relative to in control plantation forest (GLMM; $z = -3.2$, $P = 0.002$) and (**b**) may have affected caterpillar parasitism rates in adjacent native forest (though the effect of experimental herbivore reduction on parasitism rates was non-significant, as tested with a GLMM in which the interaction between herbivore reduction treatment and collection was removed during model selection). H is herbivore reduction treatment, C is control. Solid orange lines indicate a contrast significant at $\alpha = 0.05$. Error bars are s.e.m. In **a** and **b** $n = 8$ pairs of treatment and control sites as replicates.

predictors in a model with observed parasitism rate as the response variable, we found that the best model predicting observed parasitism contained only within-habitat expected parasitism rate as a significant predictor ($z = 2.8$, $P = 0.005$; Fig. 5c; Supplementary Table 6), which explained 19.9% of the variation in observed parasitism rate, while the whole model explained 27.4% of the variation. That is, predictions based on food-web data from only the host's own habitat (within-habitat expected parasitism rates) were sufficient to predict future parasitism rates, without needing data from the adjacent habitat as well, even though parasitoids shared with hosts in the adjacent habitat (Fig. 2) are known to move across the habitat edge in this system[21]. However, cross-habitat expected parasitism rate also significantly predicted observed parasitism rate when it was the only predictor in the model ($z = 2.1$, $P = 0.040$; Fig. 5d, Supplementary Table 7), wherein it explained 13.6% of the variation in observed parasitism rate. Cross-habitat expected parasitism rate was not retained in the best model as a predictor of observed parasitism rate because it explained the same component of variation in observed parasitism rate as within-habitat expected parasitism rate (Supplementary Table 6). Importantly, this collinearity between cross-habitat and within-habitat expected parasitism rate suggests that either within- or cross-habitat expected parasitism rate could be used with almost equivalent predictive success, and that measuring both may be unnecessary. In all analyses, the correlations between expected and observed parasitism rates were positive, suggesting that overall, the indirect effects occurring were apparent competition rather than apparent mutualism (which is a predator-mediated positive relationship between abundances of host species pairs[15]).

**Predictions based on binary versus quantitative food-web data.** These accurate predictions required quantitative food-web data to predict the occurrence and level of parasitism, though even with

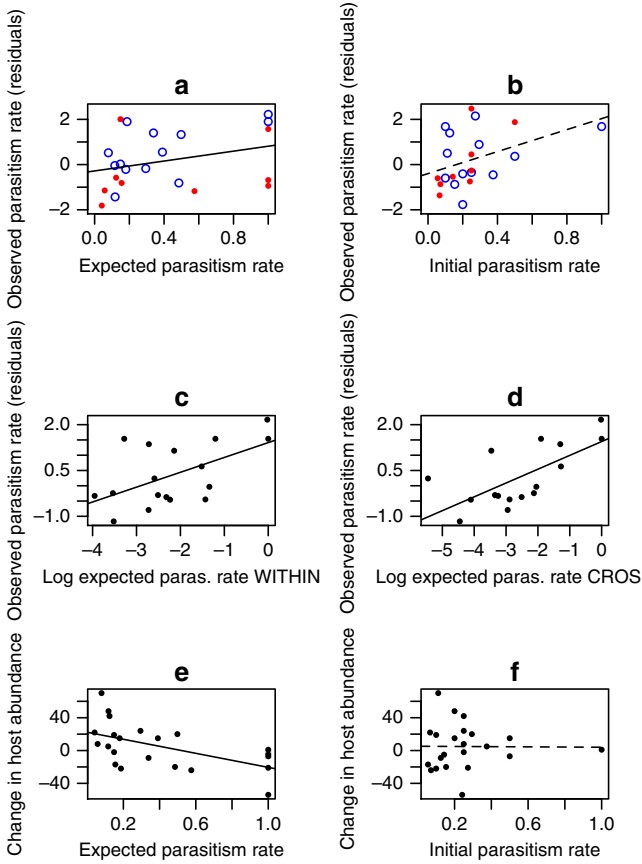

**Figure 5 | Predictions of parasitism rate and change in host abundance.**
(**a**) Expected parasitism rate predicted observed parasitism rate (GLMM; $z = 2.7$, $P = 0.007$, $R^2_{(GLMM(m))} = 0.16$), whereas (**b**) initial parasitism rate did not predict observed parasitism rate (GLMM; $z = 1.1$, $P = 0.266$). Observed parasitism rate was also significantly predicted by expected parasitism rates calculated with only (**c**) data from within the same forest as the focal host (GLMM; $z = 2.8$, $P = 0.005$, $R^2_{GLMM(m)} = 0.20$), or (**d**) data from the forest adjacent to the focal host's habitat (GLMM; $z = 2.1$, $P = 0.040$, $R^2_{(GLMM(m))} = 0.14$). (**e**) Expected parasitism rate also significantly predicted change in host abundance (linear model; $z$-value $= -3.0$, $P = 0.007$, $R^2 = 0.31$), but (**f**) initial parasitism rate did not predict change in host abundance (linear model; initial parasitism rate was removed as a predictor during model selection). In **a–d**, residuals of the best model excluding the predictor on the $x$ axis are plotted, and raw data are plotted in **e** and **f**. Residuals are deviations of the logit-linked data from the best model, and show the unexplained variation in the data remaining once variation due to the other fixed and random effects in the model have been accounted for. That is, they show the variation in the data that we are hoping to explain with the predictor on the $x$ axis. Each point represents a species within a site that was collected and successfully reared at both time steps, and was parasitized in the first time step (see equation (3); Fig. 1i). In **a** and **b** closed circles (red) represent hosts in plantation forest and open circles (blue) represent hosts in native forest, though only one fitted line is presented because there was no significant habitat$_A$ × expected parasitism rate interaction (in **a** the interaction term was removed during model selection; (**b**) interaction $z = 1.9$, $P = 0.060$). In all panels, $n = 8$ pairs of treatment and control sites as replicates.

binary information on the presence/absence of species links, some qualitative predictions could still likely be made. Expected parasitism based on a binary regional food web was a marginally non-significant predictor of observed parasitism ($z = 1.9$, $P = 0.056$, with a partial regression coefficient of 1.9 (s.e. $= 0.99$) for binary data (Supplementary Table 8a), compared with 2.3

(s.e. $= 0.84$) for quantitative data (Supplementary Table 3). When only cases of observed (non-zero) parasitism were included, binary information slightly improved predictions over those made from quantitative data (Supplementary Table 8b,c compared with Supplementary Tables 6 and 7), but since real-world applications would likely require predictions of occurrence of parasitism as well as of the level of parasitism when parasitism occurred, our results suggest that quantitative data will yield more accurate predictions. The moderate 'success' of the binary predictions suggests that apparent competitive effects were not driven only by the abundance of species, but rather by highly connected species that shared parasitoids with many others. Fortunately, these key species can be easily identified in even binary food webs.

**Apparent competition predicts changes in host abundance.** Apparent competition entails a change in the abundance of two prey species in response to altered attack rates by a shared enemy[9]. Above, we focus on attack rates rather than any resulting changes in host abundance, because attack rates measure the parasitoid-mediated top–down effects on host dynamics, without any confounding effects driven by resource competition among hosts, attack by generalist predators, or other environmental drivers. Changes in attack rates (when coupled with changes in growth rates) also form the basis of models of apparent competition[9]. However, if apparent competition is an important community-wide structuring force relative to other trophic and non-trophic direct and indirect effects on the species within a community, our expected parasitism rates should be able to predict not only observed parasitism rates, but also changes in focal host abundance across host species in the community. We found that expected parasitism rate predicted 31.1% of the variation in the change in host abundance between time steps ($z = -3.0$, $P = 0.007$; Fig. 5e; Supplementary Table 9), and that this was the same for hosts in native or plantation forest (the expected parasitism rate by host habitat interaction was removed during model selection). Predictive ability was also the same for large and small changes in abundance of other hosts within the community (the interaction between expected parasitism rate and herbivore reduction treatment was removed during model selection; Supplementary Table 9). In fact, expected parasitism rate predicted changes in host abundance better than it predicted observed parasitism rates. The better prediction of changes in host abundance than of parasitism rates could have been because the time interval over which sampling was conducted meant that we detected the effects of parasitism on abundances for more species than we detected the precise window of parasitism. Again, simply knowing host species' initial parasitism rates was not sufficient to predict their changes in abundance (Fig. 5f; Supplementary Table 10), but rather the predictions based on apparent competition were necessary.

**Summary of most important results.** We found that for each host species in the community, we could predict both the final parasitism rate, and with even greater accuracy, the change in abundance between time steps for that species. This prediction required information on the initial parasitism rate for that species, the 'normal' extent to which that host species shares parasitoids with all other host species in the region, plus the change in abundance between time steps of other host species. These predictions could equally be made for hosts in either native forest or adjacent exotic plantation forest, and for large or small changes in abundances of other host species in the system. These predictions even held when they were made for hosts in one habitat, but based only on data from the other habitat, regardless of whether the host's habitat was native or plantation forest.

## Discussion

Our finding that apparent competition is a structuring force strong enough to determine up to 31% of the variation in forest herbivore abundances has great potential utility in biological control and invasive species management, which often face poor predictive ability due to indirect effects among species[26]. Both these fields face the challenge of predicting potential impacts of introduced species or control agents before their arrival or release[30,31]. Our results show that apparent competition is an important structuring force within host–parasitoid assemblages, and that predictive models could be improved by including apparent competitive effects. Although producing our expected parasitism rates required an enormous data collection effort, which would be impractical for some applied situations, our results suggest that Muller *et al.*'s index[22] of the potential for apparent competition between pairs of host species, $d_{ij}$ (or our cross-habitat version, $d_{iAjB}$), expresses a real measure of the potential indirect trophic linkage between any two species. We show that Muller *et al.*'s index can be calculated from even regional quantitative food-web data, pooled over months and years, and still have predictive value. Thus, the shared parasitoid network of target and non-target hosts is a good proxy for predicting non-target effects of introduced biocontrol agents, even in an adjacent natural habitat or when information on host sharing comes from a different location.

Spill over of parasitoid assemblages can be greater from highly productive managed habitats to native habitats than in the opposite direction[20,21], due to bottom-up subsidies of abundant herbivores. However, we found that predicted cross-edge indirect effects did not differ significantly in magnitude (Fig. 6; Supplementary Table 11), though there was a non-significant trend towards hosts in plantation forest having stronger apparent competitive effects on hosts in native forest than vice versa, as might be expected based on the higher relative abundance of

herbivores in plantations[21]. The strength of apparent competition between habitats depended on herbivore abundance and attack rate, so higher intensity production systems with higher herbivore densities (for example, of pests) will tend to have the strongest apparent competitive effects on neighbouring habitats. Such effects would be exacerbated by land-sparing scenarios, in which production systems are intensively managed for maximum productivity, but are potentially adjacent to natural areas[32]. These impacts would be particularly likely when the edge between habitats has a low structural contrast (such as between our two forest types) that does not impede consumer movement[33], and when species overlap among habitats is high, as was the case in this system (Fig. 2). This stronger cross-habitat apparent competition between hosts when the habitats and their species are more similar poses a risk to using buffer zones of similarly structured habitats, such as plantations, around conservation areas[34], because indirect effects on native communities are likely to be greatest when more natural enemy species are shared with productive systems (that is, when the habitats have high species overlap). Host and parasitoid[21] species composition were indistinguishable (Supplementary Fig. 3a,b), and food-web structure was very similar (Supplementary Table 12) between our plantation and native forest edges. Although apparent competitive effects between habitats will be higher when the two habitats have high species similarity, predictions should work even when there are fewer shared species across the edge, such as at an agroecosystem-forest edge, because the level of regional parasitoid overlap is incorporated into the expected parasitism rate (see Methods equation (3); Fig. 1f). Nevertheless, the cross-habitat apparent competitive effects that we found are likely an edge effect rather than a coupling of interior habitat populations. Cross-habitat apparent competition is mediated by mobile predators, and cross-habitat predator subsidies are likely to decrease with distance from the habitat edge towards the interior. That said, modern landscapes are often fine-scale mosaics of different land-use types, with a proliferation of habitat edges that give prominence at a landscape scale to edge effects.

Our finding that the signal of apparent competition could be detected across an entire community, and even across habitats, is significant for three main reasons. First, apparent competition has previously been posited to structure host–parasitoid assemblages[11], and our results demonstrate empirically that sharing of parasitoids can determine attack rates on hosts and changes in host abundances across an entire network. Second, we show that network-wide predictions of attack rates and changes in host abundances can be made accurately enough to succeed even though the time step across which we made predictions would not have matched the biology of all species involved, and we used a regional rather than site-specific assessment of shared parasitism. Most importantly, our results demonstrate for the first time that entire food webs of natural habitats in production landscapes are subjected to indirect effects from adjacent production areas, such that food webs in multiple habitats function as a single metaweb of direct and indirect linkages among species across the landscape.

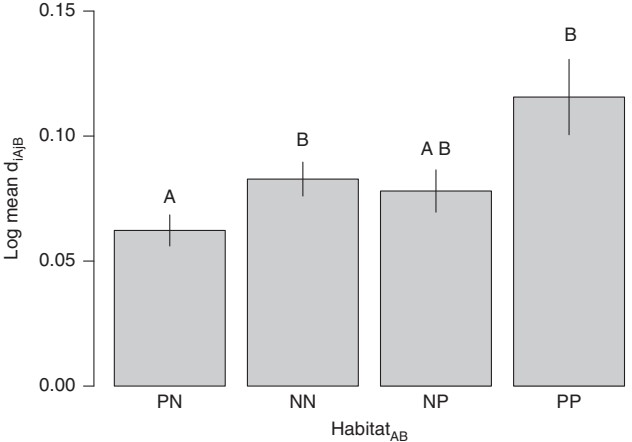

**Figure 6 | Magnitude of the potential for indirect effects ($d_{iAjB}$) among host species.** N and P refer to native forest and plantation forest, respectively, such that, for example, PN refers to the situation where host *i* (the species that is affected) is in plantation forest and host *j* (the species that generated the parasitoids attacking species *i*) is in native forest. The potential for hosts in plantation forest to affect hosts in native forest (NP) was not significantly greater than the potential for hosts in native forest to affect hosts in plantation forest (PN) through apparent competition (linear model: $t = 1.9$, $P = 0.065$). Letters above the bars are *post-hoc* mean comparisons. Error bars represent s.e.m. Data for this analysis were from the regional metaweb (Fig. 2), so sites were pooled, and habitat-specific species pairs were replicates (PN: $n = 178$; NN: $n = 255$; NP: $n = 178$; PP: $n = 155$).

## Methods

**Study sites and sampling overview.** We selected two sets of forest edge sites between plantation *Pinus radiata* and native southern beech (Nothofagaceae) forest: eight training sites, and 16 validation sites, in the Nelson/Marlborough region of South Island, New Zealand (Supplementary Fig. 4). At the training sites, we collected species interaction data to create a regional 'metaweb' (a single web made up of all the webs from separate sites; see below; Fig. 1a–c). The purpose of this metaweb was to maximize resolution of potential host–parasitoid linkages, from which to derive predictions about the potential for indirect interactions based on shared parasitoids (Fig. 1d,e). The training sites were at least 2.7 km apart, and at least 1 km from any of the 16 experimental validation sites, and interactions were

sampled seven times over two summer seasons (season 1: December 2009, January, February 2010; season 2: October, November 2010, January, February 2011; Fig. 1a).

Towards the end of the training data collection, in early 2011, we experimentally reduced herbivore abundance on the plantation side of the edge in half of our validation sites (at eight herbivore reduction sites), and each of these was paired with a control site, within eight spatial blocks (Fig. 1h). Spatial blocks were at least 2.7 km apart, and within each block the pair of validation edge sites (one herbivore reduction site paired with one control site) was at least 1 km apart, but not more than 2.7 km apart. Validation sites were sampled twice before the herbivore reduction treatment (October, November 2010; Fig. 1g) and twice after the herbivore reduction treatment (January, February 2011; Fig. 1h), in a BACI (before–after controlled intervention) design. The pine forests at our edges were between 17 and 28 years of age, with trees mature enough that the canopies were closed. We describe the understory vegetation composition of the two forest types elsewhere[21,35]. Each forest (plantation or native) was large enough to have an interior location at least 400 m from all edges.

### Herbivore reduction at half of the validation sites.
Herbivore abundances fluctuate through time, though occasionally there are large changes in abundance following harvesting, pesticide application or other events such as disease spread or climatic extremes. To ensure that our results would be applicable to both large and small changes in abundance, we generated large changes in herbivore abundance on the plantation side of the edge at each herbivore reduction site, with the expectation that this would reduce the *in situ* production of parasitoids and thereby affect parasitism rates in the plantation, and possibly in the adjacent native forest. We applied an herbivore reduction treatment to one of the validation sites in each spatial block, leaving the other as a control. We timed our herbivore reduction experiment to be roughly between parasitoid larval generations, such that most parasitoids present would be in adult form, and thus not killed inside hosts during the herbivore reduction spray (Supplementary Fig. 1).

On 30 December 2010 and again on 9 January 2011, we sprayed the plantation forest at each herbivore reduction site with Delfin WG (Certis, USA L.L.C.), a commercial formulation of *Bacillus thuringiensis* var. *kurstaki*—an organic, non-persistent pesticide. This bacterial strain kills larval Lepidoptera (our hosts) upon ingestion, but does not affect other insects. We sprayed an area of 2.5 ha at each herbivore reduction site (250 m along the edge, with the sampling transect at the centre, by 100 m into the interior of the pine forest), using a helicopter with micron air nozzles (droplet size ~100 microns). In each spray run we applied 4.5 kg ha$^{-1}$ of Delfin WG, mixed with 0.125 l ha$^{-1}$ of the wetting agent Du-Wet (Elliot Chemicals Ltd., Auckland, NZ), and water according to aerial spray guidelines. These amounts and timing were according to the manufacturers' instructions for maximal effectiveness, and comparable to amounts found to be maximally effective against lepidopteran pests (Tortricidae) in North American coniferous forests[36].

We used transect data on caterpillar abundances before and after the herbivore reduction treatment (see next section) to test whether the treatment had the desired effect. We fitted a generalized linear mixed model with a Poisson error distribution (including the canonical log link function), and caterpillar abundance in plantation forest as the response variable. The predictors were collection (two levels: with the two before-herbivore-reduction collections pooled into one 'before' sample versus the two after-herbivore-reduction collections pooled into one 'after' sample), treatment (herbivore reduction versus control) and the collection x treatment interaction as fixed effects. Forest type, nested within site, nested within block were included as random factors to account for the non-independence of repeated measures at the before and after time steps. As desired, the spray significantly reduced caterpillar abundance in the herbivore reduction treatments relative to the control treatments (interaction effect $z = -3.18$, $P = 0.002$, Fig. 4a, light green bars; Supplementary Table 1).

### Sampling of species and interactions.
Sampling procedures were the same at training and validation sites (Fig. 1a,g,h). To collect quantitative food-web data from which to assess indirect species interactions, we sampled lepidopteran larvae (caterpillars) and their parasitoids from the adjacent native and plantation forests at each site. In each sampling round, we collected caterpillars at each site by establishing one 50 m transect in each forest type, 10 m from and parallel to the edge, which we designated as the last row of pine trees abutting the native forest. The 10 m distance was a compromise between a small spatial scale over which parasitoids would be likely to disperse[37], and a distance from the edge great enough that the vegetation was distinct from that of the adjacent habitat.

We sampled all vegetation within 1 m on either side of each transect, up to a height of 2 m, by beating each plant and holding a sheet underneath to catch all of the caterpillars that were dislodged. At ten points (that is, at 5 m intervals along each transect), we sampled the lower canopy up to a height of 9 m, within an area of 1 m$^2$, by clipping all vegetation using an extendable pole with a clipper head on the end. If canopy foliage was not reachable at a designated clipping point, we clipped four or five branches (an approximately standard number of leaves) from the nearest reachable point. We beat all the clippings over large sheets and collected the caterpillars as for the understory samples. When transect sampling yielded fewer than 50 individuals, we carried out extra sampling of vegetation on the

non-edge side of the transect, and as near to the transect as possible, until either 50 caterpillars were found, or we had sampled for two person hours. We used these extra sampling caterpillars to obtain more accurate estimates of per capita parasitism rate and to identify parasitoid–host interactions, but did not include them in any herbivore abundance calculations.

We housed the collected caterpillars individually, and reared each to adulthood or parasitoid emergence by feeding it foliage of the host plant on which it had been found, supplemented with artificial diet formulated for Beet Army Worm (Noctuidae; Bio-Serv Entomology Custom Research Diets and Environmental Enrichment Products, New Jersey, USA). After emergence, parasitoids and moths were identified to species level where possible using available taxonomic information[38–41] and expert assistance (see Acknowledgements), and otherwise to morphospecies (hereafter species; Fig. 1b,g,h).

Congeneric lepidopteran species that were indistinguishable as larvae were lumped, because when parasitoids emerged, larval morphology was our only means of identifying the host. However, there were only 11 taxa lumped in this way, and these should not have affected the results of the study other than to create a more conservative assessment of whether indirect effects are important in structuring communities. The assessment would have been more conservative because erroneously lumping two distinct species in this analysis would label some of the interspecific indirect effects on either of the lumped species as intraspecific effects. In part of our analysis we removed the contribution of intraspecific effects to the expected parasitism rates, so this lumping of species would only have made us less likely to find that expected parasitism rates significantly predicted observed parasitism rates.

We found parasitism by Hymenoptera, Diptera (Tachinidae) and Nematoda (Mermithidae). Although little is known about the dispersal capabilities of Nematoda relative to Hymenoptera and Diptera, there is evidence that Nematoda can disperse long distances by phoresis[42], by infected adults[43] and by wind[44]. These nematodes also killed the host before emerging, and thus could be accurately described as parasitoids. Therefore, we included parasitism events by Nematoda in our analysis, in order to consider the most complete host–parasitoid assemblage possible rather than an arbitrary subset of this guild based on taxonomy.

### Parasitoid molecular identification.
Most of the parasitoid wasps collected were of undescribed species. Therefore, in order to match males with females (which may have different morphology), we DNA-barcoded representative female specimens that covered the observed morphological variation within each morphospecies, and all male specimens, in order to link these genetically to the female taxa. To identify nematode parasitoids we used an exclusively molecular technique.

To molecularly match male and female parasitoid Hymenoptera and to check morphospecies, we amplified and sequenced the Cytochrome c oxidase subunit I (COI) region of the mitochondrial DNA. We removed a leg from each wasp, crushed it, and performed a DNA extraction using the prepGEM Insect kit (ZyGEM Corporation, New Zealand). The extracted DNA was used as a template to amplify the target COI region using the primer pair HCO2198 (Folmer): 5′-TAA ACT TCA GGG TGA CCA AAA AAT CA-3′ and LCO1490 (Folmer): 5′-GGT CAA CAA ATC ATA AAG ATA TTG G-3′ (ref. 45), with KAPA Blood PCR Kit (Kapa Biosystems, USA). We used 50 μl reactions and the following thermal cycling conditions: 94 °C for 3 min, 30 cycles of 94 °C (30 s), 52 °C (30 s), 72 °C (40 s), and a final extension of 72 °C for 2 min. 10 μl of the amplified product was resolved on a 1% agarose gel stained with SYBR Safe DNA Gel Stain (Life Technologies, USA) to check for correct amplification of the ~650 bp product. The amplicons were purified using *GenCatch* PCR Purification Kit (Epoch Life Science, USA), and the purified product was Sanger sequenced at Macrogen (Seoul, Korea). The resulting sequence reads were checked and edited using *MEGA*[46] version 5.

We then used Sequence Demarcation Tool[47] v.1.2 to calculate pairwise similarity for each pair of aligned sequences, and used MUSCLE[48] to re-align the sequences and cluster them based on similarity scores, using a rooted neighbour-joining tree. We used a matrix of species-by-species similarity scores[47] to match unidentified males to female morphospecies, and to lump within genera those morphospecies that sequence similarity suggested should be considered a single species. We did not set a strict per cent similarity species demarcation criterion because we did not sequence all of our specimens, and therefore we could not use molecular information to split morphospecies. Rather, we lumped morphospecies based on obvious per cent similarity groupings in the species-by-species similarity matrix. The lowest per cent similarity between lumped morphospecies was 96.05%. All insect parasitoid specimens have been deposited at the New Zealand Arthropod Collection, Auckland, NZ (Tachinidae and Hymenoptera other than Braconidae), and the Museum of New Zealand Te Papa Tongarewa, Wellington, NZ (Braconidae), and the *COI* sequences of the barcoded wasps have been uploaded onto GenBank (KM106851—KM107193).

To identify all individual nematode parasitoids, we used an exclusively molecular technique. The total DNA was extracted using the Extract-N-Amp Tissue PCR Kit (Sigma-Aldrich, USA). We amplified a ~1 kb region of the 18S ribosomal RNA sequence, using the primer pair Nem_18S_F: 5′-CGC GAA TRG CTC ATT ACA ACA GC-3′ and Nem_18S_R: 5′-GGG CGG TAT CTG ATC GCC-3′ (ref. 49), with KAPA Blood PCR Kit (Kapa Biosystems, USA). We used the following thermal cycling conditions: 94 °C for 3 min, 30 cycles of 94 °C (30 s),

55 °C (30 s), 72 °C (1 min), and a final extension of 72 °C for 2 min. We used the same purification, sequencing, and analysis methods for the nematode amplicons as for the Hymenoptera amplicons. The GenBank accession numbers for the nematode sequences are KP307028—KP307059.

**Calculating a regional measure of shared parasitism.** To generate a regional quantitative measure of shared parasitoids for each host species pair in the region, with maximum possible resolution, we created a quantitative food web for the region (the 'metaweb' of pooled data from our training sites, Fig. 1b,c). We then used this regional metaweb to calculate a quantitative measure of shared parasitoids, or the potential for apparent competition, $d_{ij}$ (ref. 22; the dependence of parasitoids of host species $i$ on host species $j$; Fig. 1d,e). This dependence, $d_{ij}$, measures the proportion of parasitoids attacking host species $i$ that recruited from host species $j$, for every pair of host species in a community[22]:

$$d_{ij} = \sum_{k=1}^{P} \left[ \frac{\alpha_{ik}}{\sum_{l=1}^{P} \alpha_{il}} \frac{\alpha_{jk}}{\sum_{m=1}^{H} \alpha_{mk}} \right] \qquad (1)$$

where $\alpha$ is the link strength (that is, number of attacks), $i$ and $j$ are a focal host species pair, $m$ is all host species from 1 to $H$ (the total number of host species), $k$ is a parasitoid species, and $l$ is all parasitoid species, from 1 to $P$ (the total number of parasitoid species).

However, to extend this equation to multiple habitats, we indexed host species by habitat (Fig. 2c) such that, for example, species $i$ in habitat $A$ (for example, plantation forest) would be treated as a separate species from species $i$ in habitat $B$ (for example, native forest) in equation (1). Effectively, this expanded equation (1) to explicitly consider two habitats, each containing hosts that share parasitoids, which move freely between habitats. Although it is not possible to know the extent to which each parasitoid individual views the native/plantation edge as a boundary, our previous work in this region found considerable movement of adults of our parasitoid species in both directions across the edge[21]. Furthermore, we incorporated habitat effects into our hypothesis tests to determine whether this assumption was violated. Thus, we calculated $d_{iAjB}$, the habitat-specific contribution to parasitism of host $i$ by parasitoids of host $j$:

$$d_{iAjB} = \sum_{k=1}^{P} \left[ \frac{\alpha_{iAk}}{\sum_{l=1}^{P} \alpha_{iAl}} \frac{\alpha_{jBk}}{\sum_{m=1}^{H_q} \alpha_{mk}} \right] \qquad (2)$$

where $d_{iAjB}$ is the proportion of parasitoids attacking species $i$ in habitat $A$ that were reared from species $j$ in habitat $B$. $A$ is the habitat of host species $i$, $B$ is the habitat of host species $j$, $H_q$ is the total number of host species from the total pool of $q$ habitats producing parasitoids. For all calculations with equation (2) in this study, $q$ includes both the native and plantation forests. All other variables are as defined in equation (1).

The first part of equation (2) represents the fraction of attacks by parasitoid species $k$ on host species $i$ in habitat $A$ out of the total number of attacks by all of the $P$ species of parasitoid on host $i$ in habitat $A$. This is then multiplied by the number of parasitoids of species $k$ that were reared out of host species $j$ in habitat $B$ during the same sampling period, divided by the total number of individuals of parasitoid species $k$ that were reared out of all of the $H$ host species in either of the habitats considered.

In the case where $A = B$, $d_{iAjB}$ measures within-habitat shared parasitism, and when $A \neq B$, $d_{iAjB}$ measures cross-habitat shared parasitism. However, in both of these cases $q = 2$, since even for parasitism within only one of the habitats, both habitats will contribute to the total pool of parasitoids. If the total pool of parasitoids occurs in only one habitat, equation (2) simplifies to equation (1) (when $A = B$ and $q = 1$). When $i = j$, $d_{iAjB}$ measures the proportion of parasitoids attacking species $i$ that recruit from species $i$ in the same ($A = B$) or different ($A \neq B$) habitat (that is, the intraspecific contribution of $i$ to its own parasitoid pool).

**Expected, observed and initial parasitism rates.** We tested whether we could use knowledge of the proportions of shared parasitoids ($d_{iAjB}$ values) between host species in our training metaweb (Fig. 1e), as well as initial attack rates from (pre-treatment) time $t$ at our validation sites (Fig. 1g), to predict parasitism rates at (post-treatment) time $t + 1$ at the validation sites (Fig. 1h), given known changes in host abundances (Fig. 1g,h). Parasitism rate refers to the number of parasitism events divided by the number of hosts sampled. We first calculated the expected parasitism rate at time $t + 1$ of host species $i$ in habitat $A$ using:

$$E_{iA(t+1)} = \sum_{j=1}^{H_q} \left( \frac{d_{iAjB} \sum_{l=1}^{P} \alpha_{iAlt}}{n_{jBt}} n_{jB(t+1)} \right) \left( \frac{1}{n_{iA(t+1)}} \right) \qquad (3)$$

where $n$ is host abundance, $t$ is a time step (before or after sampling dates), and all other variables are defined as in equation (2). Here, when $q = 2$, $B$ can take values of either habitat. When $q = 1$, $B$ is limited to being either one habitat or the other (here plantation or native forest). That is, when $q = 2$, this equation calculates the expected parasitism rate of host $i$ in habitat $A$ based on potential apparent competition with hosts in the same and the adjacent habitat. When $q = 1$ it calculates the expected parasitism rate of host $i$ in habitat $A$ based on potential apparent competition with hosts either in plantation or in native forest. In both cases, $A$ can

take values of either habitat. We calculated $d_{iAjB}$ from the metaweb (transect plus extra sampling data for maximum resolution on interactions) from our training sites (Fig. 1d,e), and $\alpha_{iAl(t)}$ (transect plus extra sampling data) and $n_{jB(t)}$ (transect data only, for a standardized measure of abundance) from our validation sites in the pre-herbivore-reduction (time $t$) samples (in both reduction and control sites; Fig. 1g). We calculated $n_{jB(t+1)}$ (transect data only) and $n_{iA(t+1)}$ (transect plus extra sampling data, since $\alpha_{iAl(t)}$ was calculated from transect plus extra sampling data) from the post-herbivore-reduction samples (also in both reduction and control sites; Fig. 1h). Equation (3) calculates, for every host $j$ (in either habitat) that shares parasitoids with host $i$ in habitat $A$, the expected per capita attack rate on host $i$ of parasitoids that were reared from host $j$. This expected per capita attack rate on host $i$ due to parasitoids from host $j$ is then multiplied by the $t + 1$ abundance of species $j$ to convert the rate to a number of parasitism events. When summed over all $H$, this gives the expected number of attacks (not rate) on species $i$ at time $t + 1$, which can then be divided by the abundance of species $i$ at time $t + 1$ to give the expected parasitism rate (Fig. 1i). In cases where more attacks were predicted than hosts were collected in the $t + 1$ collection, expected parasitism rates were greater than one. We reduced these expected parasitism rates to 1 in our analysis because our definition of parasitism rate was parasitized hosts/total hosts (that is, host centric), so rates greater than one are not possible.

Equation (3) is written such that the expected parasitism rate of host $i$ increases in proportion to the abundance of host $j$. Thus, it assumes apparent competition between hosts that share parasitoids, rather than apparent mutualism. A positive correlation between expected and observed parasitism rate would therefore suggest that apparent competition is more important than apparent mutualism, whereas a negative correlation would suggest that apparent mutualism is more important than competition.

For the entire herbivore assemblage, we tested whether this expected parasitism rate significantly predicted the observed parasitism rate of each host species $i$ at time $t + 1$, which was calculated as:

$$O_{iA(t+1)} = \frac{\sum_{l=1}^{P} \alpha_{iAl(t+1)}}{n_{iA(t+1)}} \qquad (4)$$

where all variables are defined and calculated as in equation (3) (Fig. 1j).

To test whether predictions based on apparent competition were necessary to predict observed parasitism rates, we also tested whether observed parasitism rates could be predicted simply by using initial parasitism rates:

$$I_{iAt} = \frac{\alpha_{iAlt}}{n_{iAt}} \qquad (5)$$

where all variables are defined as above and both $\alpha_{iAl(t)}$ and $n_{iAt}$ are calculated from transect plus extra sampling data.

Finally, because apparent competition is defined as a change in abundance, rather than parasitism rate, of one species as a result of an indirect interaction with another species via a shared predator, we tested whether our expected parasitism rate for all species $i$ could predict the change in abundance between time steps for all species $i$ (Fig. 1k). Change in abundance was calculated as:

$$\Delta n_{iA} = n_{iA(t+1)} - n_{iAt} \qquad (6)$$

where all variables are defined as above and $n_{iA(t+1)}$ and $n_{iAt}$ are calculated from transect data only.

**Hypothesis testing.** We first tested whether expected parasitism rate based on shared parasitoids from both habitats could predict observed parasitism rate (Fig. 1l). This test was to determine whether parasitoid-mediated indirect effects structure attack rates across the entire host–parasitoid assemblage to the extent that quantitative food webs can be used to predict parasitism rates. Unlike in previous studies[15,27,50], these hosts were not chosen based on predicted strength of interactions, but rather included all the hosts that were attacked at time $t + 1$. This test assumed that one pool of parasitoids was shared between the two adjacent habitats, with a habitat term included in the model to determine whether violation of this assumption was masking a relationship between expected and observed parasitism rate. The interaction between the habitat term and expected parasitism rate tested whether predictive power of expected parasitism rate depended on the habitat of host $i$, since the habitat edge might filter natural enemies in one direction[33], and therefore make parasitism rates in one habitat less predictable because of the cross-habitat contribution to the expected parasitism rate.

We used a generalized linear mixed model (GLMM) with a binomial distribution, in which $O_{iA(t+1)}$ (equation (4)) was predicted by $E_{iA(t+1)}$ (equation (3)), habitat$_A$, herbivore reduction treatment, and all interactions, with forest nested within site nested within block included as random factors. A significant three-way $E_{iA(t+1)} \times$ habitat$_A \times$ herbivore reduction treatment interaction would mean that expected parasitism rate predicts observed parasitism rate with different success depending on the habitat of the focal host, and that predictive ability within habitat also depends on the magnitude of change in host abundance.

In all analyses, to select the best model we first selected the optimal random factor structure[51]. We ran the full model, as well as models with all combinations of nested random factors or each random factor singly, and selected the model with the lowest Akaike Information Criterion (AIC) value[52]. Then, using this optimal

random factor structure, we selected the best-fitting fixed effects structure, again by running the full model as well as all possible simpler models, and selecting the one with the lowest AIC value. Where models differed by <2 AIC points, we selected the simpler model to adhere to the principle of parsimony. All generalized linear mixed models were tested in the lme4 package[53] in R version 3.1.2 (ref. 54). Here and in all subsequent analyses we tested for overdispersion, and found that models were not overdispersed (that is, the ratio of the sum of squared Pearson residuals to residual degrees of freedom was <2 and non-significant when tested with a Chi-squared test.). To calculate the per cent of variation explained by expected parasitism rate in this and all subsequent analyses using mixed effects models, we calculated Nakagawa and Schielzeth's marginal $R^2$ ($R^2_{GLMM(m)}$)[55]. We also calculated conditional $R^2$ ($R^2_{GLMM(c)}$) to look at the variation explained by the fixed and random effects. For linear models we report multiple $R^2$. To test how the inclusion of each fixed effect changed the variance components at different levels of each mixed model, we calculated the per cent change in variance for each level (PCV) by comparing the final model with a null (intercept only) model that had the same random effects structure[55].

We were interested in whether the contribution to expected parasitism rate by interspecific and cross-habitat intraspecific indirect effects could predict observed parasitism rates. To test this, we repeated the above hypothesis tests, this time excluding within-habitat intraspecific contributions (that is, delayed density-dependent parasitism, where $i = j$ and $A = B$) to the expected parasitism rate calculation. To test whether predictions based on apparent competition could better predict observed parasitism rate than could initial parasitism rate, we repeated the above hypothesis tests, but with initial parasitism rate as the predictor variable.

To better understand our initial results, we tested whether predicted cross-habitat indirect effects were realized, or whether the predictive ability of expected parasitism rate is largely due to its prediction of within-habitat effects. For this analysis we included only cases in which parasitism was observed at $t + 1$, because zero values (no parasitism) could not be attributed to either within- or cross-habitat hosts. We separated the expected parasitism rates into the expected attacks based on cross-habitat versus within-habitat host abundances. To calculate cross-habitat versus within-habitat expected parasitism rates, we set $q$ equal to 1 in equation (3), and first set $B$ (the habitat of host $j$) to be plantation forest. These settings allowed us to calculate expected parasitism rates for host $i$ in both plantation and native forest based on apparent competition with hosts in plantation forest. For host $i$ in plantation forest, this would produce within-habitat expected parasitism rates, and for host $i$ in native forest, this would produce cross-habitat expected parasitism rates. We then set $B$ to be native forest, and again calculated $E_{iA(t+1)}$ for host $i$. For host $i$ in plantation forest, this produced cross-habitat expected parasitism rates, and for host $i$ in native forest, this produced within-habitat expected parasitism rates. This process produced both within- and cross-habitat expected parasitism rates for each host species $i$ in plantation forest, and each host species $i$ in native forest. To test whether cross-habitat expected parasitism rates predicted observed parasitism rate, and whether they predicted a component of the variation in observed parasitism rate beyond that predicted by within-habitat expected parasitism rate, we used a GLMM with a binomial distribution and $O_{iA(t+1)}$ as the response variable. Within-habitat and cross-habitat expected parasitism rates were included as separate fixed predictors, and forest within site within block were included as nested random factors in the full model, before selection of the best random and then fixed effect structure. Expected parasitism rates were log-transformed to achieve a more even distribution of values and improve linear model fit (but results were not qualitatively different without this transformation; data not shown). Hypothesis tests were based on partial coefficients, so if both fixed effects were retained in the final model, a significant effect of the cross-habitat term would indicate additional explanatory power beyond that of within-habitat expectation or any collinearity among the two predictors. $R^2_{GLMM(m)}$, $R^2_{GLMM(c)}$ and PCV were calculated as above.

To test whether expected parasitism rate could predict change in host abundance, we used a linear mixed effects model with change in host abundance, $\Delta n_{iA}$, as the response variable, $E_{iA(t+1)}$, Habitat$_A$, and herbivore reduction treatment as fixed predictor variables, and forest nested within site within block as random factors. We checked assumptions of normality and homogeneity of variance, and since during our random factor selection procedure, we found that the random factors did not account for any of the variance in change in host abundance, in the end we used a linear model. We also tested whether initial parasitism rate could predict change in host abundance using a linear model with $\Delta n_{iA}$, as the response variable and $I_{iAt}$, Habitat$_A$, and herbivore reduction treatment as fixed predictor variables.

A significant interaction effect between habitat$_A$ and expected parasitism rate in the first model described would mean that cross-habitat indirect effects occur asymmetrically (across habitat types) in the number of expected interactions that are realized. However, we also tested whether the magnitude of predicted indirect effects was stronger in one direction or the other (based on previous results of asymmetric flow of parasitoids from plantation to native forest[21]), and compared this to the within-habitat strengths of predicted indirect effects. To do this we tested whether $d_{iAjB}$ was significantly different for host pairs depending on the habitats of the indirectly 'affected' (species $i$) versus 'affecting' (species $j$) hosts (habitat$_{AB}$). For this test, we used data from the metaweb sampled at training sites,

since this was sampled over the largest time period, and would therefore include the largest sample of potential apparent competitive/mutualistic linkages. We used an ANOVA to test whether the magnitude of log $d_{iAjB}$ was related to the identities of $A$ and $B$ in habitat$_{AB}$, where habitat$_{AB}$ could be PN (Pine-Native), NN, NP, or PP. (For example, 'PN' refers to the situation where host $i$, the host affected by apparent competition, is in plantation forest and host $j$, the host causing apparent competitive effects, is in native forest.)

**Data availability.** Computer code for all analyses presented here is available from the authors upon request. All relevant data are also available from the authors upon request. The GenBank accession numbers for the nematode sequences are KP307028—KP307059. All insect parasitoid specimens have been deposited at the New Zealand Arthropod Collection, Auckland, NZ (Tachinidae and Hymenoptera other than Braconidae), and the Museum of New Zealand Te Papa Tongarewa, Wellington, NZ (Braconidae), and the COI sequences of the barcoded wasps have been uploaded onto GenBank (KM106851—KM107193). This article is based on a doctoral thesis chapter[56].

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

## Acknowledgements

We thank Nelson Forests Ltd., Merrill & Ring, Hancock Timber Resource Group, M. Turbitt, D. Bryant, N. Buchanan, L. and P. Douglas, and the Department of Conservation for forest access. J. Dugdale, J. Berry, and R. Schnitzler provided taxonomic assistance. Members of the Ladley family (J., B., D., D., and S.) D. Conder, N. Etheridge, and D. Payton assisted with field and lab logistics. Y. Brindle, C. Hohe, S. Litchwark, S. Hunt, A. McLeod, L. O'Brien, A. Knight, L. Williamson, T. Lambert, H. McFarland, E. Allen, C. Thomas, R. McGee, K. Trotter, T. Watson, V. Nguyen, A. Young, D. Davies, and M. Bartlett assisted with caterpillar collection and rearing. The Tylianakis and Stouffer labs, J. Beggs, H.C.J. Godfray, K.S. McCann, and T. Roslin provided comments on the manuscript. C.M.F. was supported by the Natural Sciences and Engineering Research Council of Canada, Education New Zealand, and the University of Canterbury. The project itself, J.M.T and G.P. were supported by the Marsden Fund (UOC-0802). R.K.D. was funded by an Australian Research Council Future Fellowship FT100100040. A.V. is supported by the National Research Foundation of South Africa. J.M.T. is funded by a Rutherford Discovery Fellowship, administered by the Royal Society of New Zealand.

## Author contributions

C.M.F. and G.P. carried out field work and identification of samples, C.M.F. wrote the first draft of the manuscript and carried out the statistical analysis with input from J.M.T. and R.K.D. J.M.T., R.K.D. and T.A.R. secured funding and developed the idea with C.M.F. A.V. led the DNA barcoding and molecular identification of specimens, with C.M.F. and G.P. All authors commented on manuscript drafts and contributed to writing.

## Additional information

**Competing financial interests:** The authors declare no competing financial interests.

