## [Peer review file · Nature Communications]

Reviewers' Comments:

Reviewer #1 (Remarks to the Author)

A. Summary of the key results

The authors show that food web data can predict changes in the attack rates of natural enemies on hosts in neighbouring habitats, following natural and experimental changes in host abundances.

B. Originality and interest. The novelty of the manuscript comes from the experimental approach, the investigation of 'across-habitat' effects, and the fact that it is applied across a very large number of interacting species. Previous investigations (cited by the authors) have looked at across-boundary effects for small sets of sets of interacting species and/or have used observational rather than experimental methods. It seeks to address important ecological questions (Can we predict dynamics from 'static' food web patterns? Do shared enemies determine the relative abundance of species?) as well as applied questions (Can mobile, shared consumers link population dynamics between man-modified and 'natural' habitats?).

C. Data & methodology

This is a hugely impressive, large-scale dataset. The effort that has gone into compiling these data should not be underestimated! Methods of data manipulation and analysis are pretty complicated and difficult to follow - Extended Data Figure 2 was useful to summarise and explain the process and I wonder if this could be used in the main text?

D. Appropriate use of statistics and treatment of uncertainties

All the analysis is limited to parasitism rates but ultimately for apparent competition these should translate into changes in abundance. I understand the authors' logic (lines 88-94) for concentrating on parasitism rates, but why not also analyse and report effects on abundance? This would tell us whether the effects of apparent competition are actually having an impact on abundances, in the context of the 'other drivers' of host population size that the authors mention; fundamentally this is how apparent competition is defined. Surely if you want to make the case that apparent competition is important in this system (the main claim of the manuscript) then you need to test if it actually influences host abundances? Otherwise we are left thinking that its effect, though statistically significant, may be minimal - yes it explains 15.6% of variation in parasitism rates but how does this modest contribution translate into population dynamic terms, relative to those other potentially important factors influencing host abundances?

E. Conclusions: robustness, validity, reliability

Figure 3 is a key figure that seeks to encapsulate the main result but it is not hugely convincing! While I trust the authors' models and analysis skills, it would be good to have strong reassurance from plotting of data that the statistically significant patterns identified in the models are valid and are biologically as well as statistically significant. This is particularly the case when the models are complex, as here.

Figure 3 also needs better explanation! This may be my failing but I think the data points are individual host species (separated by site), according to the main text (not explained in the legend)- but why then are there so few data points? I also don't follow why the residuals vary between -2 and +2. Rates (both predicted or observed) must surely vary between 0 and 1 by definition, so how can you get a rate residual of -2?

F. Suggested improvements

It would not be realistic or helpful to compile more data to improve the manuscript. The ms could be improved with better explanation and exposition of results as described elsewhere, and by analysing the implications for host abundance as well as parasitism.

G. References

These are fine and give appropriate credit to previous work in this area.

H. Clarity and context: lucidity of abstract/summary, appropriateness of abstract, introduction and conclusions

The analytical approach taken is complex (through necessity, I think) and the reader has to work hard to understand the methods used - I really needed the Supplementary Material to get to grips with both the predicting of indirect interactions, and the modelling. I think the text could be improved and simplified to help with comprehension.

For example:

- the Abstract does not really include a clear description of what the authors have actually done/investigated e.g. make it clear that this study involves a large scale manipulative experiment so that the results stand on their own.

- Lines 95-96 - this is the key sentence to encapsulate the results but it is not easy to understand on its own. This is the first mention of a 'model' and at this point the reader doesn't really know what is being predicted from what.

Minor comment:

Line 324 Parasitism "rates" greater than one are in fact possible: superparasitism or multiparasitism, observed in many parasitoids. However their outcome won't be documented with data collected in the way the authors describe, based on successful emergence rather than dissection of hosts or sequencing of parasitoids within hosts.

Reviewer #2 (Remarks to the Author)

This manuscript is about apparent competition between Lepidoptera species due to shared parasitoids across a habitat boundary, and in response to changes in host abundance. The two habitats are plantation and natural forest. The change in host abundance is due to seasonal variation, and a one-time insecticide spray. The authors found the host and parasitoid community to be the same in the two habitats. There was strong apparent competition. The change in attack rate that occurred with change in host abundance in one habitat (and also presumably parasitoid abundance) could be predicted based on current attack rate, demonstrating that apparent competition was an important force linking what happens in one habitat with what happens in the other.

Overall it is a very interesting study, both as basic ecology, and as it applied to our understanding of the impact of cultivation in the landscape, and conservation biological control. However, I found the rationale for the study and presentation of the results unclear and in some cases misleading or oversimplified.

More detailed comments:

I have made notes on the attached PDF, and present just the main points here:

There was some information presented in the methods that should have been presented in the introduction and results. Such as a simple statement of hypotheses being tested, and what data were collected.

I strongly recommend explaining to us the connection between this ms and references 24 and 7. It is the same research system and collection style. Is it the same data? It is confusing to try to understand the apparent competition part of the story without the known information about

spillover and the match between the two communities. Also, it is hard to see what is presented here that isn't presented there (outside of the predictions of attack rate calculations which is for sure a main point of this ms).

Intro paragraph:

I found this first paragraph (mostly lines 34-41) to be not very informative about the study. I made suggestions, but my overall suggestion is to rewrite the intro in more straight forward language.

It would also be easier to understand the paragraph is paragraph might work better if the idea of flow between production and non-production was introduced earlier (now in line 50).

Line 53: Land use changes also bring in new species.

Lines 72-74: I see how the spray changes host abundance in one habitat and not the other, but the natural seasonal changes must be the same in both habitats. It isn't clear to me what the specific hypotheses are about apparent competition and changes in the host abundances

Line 77: If just edges were samples, how were the two habitats compared? (I figures out the answer to this later when I read the methods, but it should be presented here.

Lines 85-86: Wouldn't this also kill all the parasitoids in the Lep larvae? At least it would kill specific types (that were not adults at the moment), which would change the food web.

Lines 98-102: This seems strange to me because:

1) the experimental manipulation would kill leps in just the plantation (is that how it went)? and not the forest, where as the seasonal change in abundance would probably occur in both habitats.

2) The spray would kill parasitoids that are in or on leps, so the parasitoid community would be different (also lines 209-211).

Lines 156-160: I think the general applicability of the results is exaggerated here because the match between the insects in an agricultural field and surrounding landscape is generally low, and parasitoids generally have narrow host ranges. Maybe it would work better when thinking of predators.

Lines 167-169: Not sure what you mean by regional vs site specific assessment of shared parasitism.

Line 220: Why assume the parasitoids disperse a small distance? If they do, then doesn't that change the interpretation of the results quite a lot?

Line 551: Which one is habitat A?

Line 608: This figure is nice. Why hide it in the supplementary material?

Reviewer #3 (Remarks to the Author)

This MS reports on an unprecedented landscape-scale manipulation to test a foundational idea- apparent competition-in community ecology. However, I have a lot of issues with the MS. First, and perhaps most importantly, the MS is not written in a way that is even remotely accessible to the broad readership of Nature Communications. It is laden with technical jargon and incomplete explanation of concepts and rationale.

The main prediction of apparent competition is that increases in the abundance of one species can indirectly decrease the abundance of another species through changes in species consumption by a shared natural enemy. Classic apparent competition assumes all species exist in the same habitat. This experiment tests this idea using insect prey-parasitoid nature enemy system but adds an interesting twist: it looks at spillover effects of changes in prey species abundance in one habitat on prey species in and adjacent habitat. Mobile parasitoid species that shuttle between habitats are hypothesized to be the mediators. Indeed, the entire study is predicated on this key assumption of mobility. However, the study never reports data to validate this lynchpin assumption. In addition, the analysis examines interactions merely within a very short distance from the edge between 2 habitats: it doesn't test whether there is spillover to the broader extent of each habitat. Therefore, the results could be a transient edge effect rather than an explanation of process across the landscape. Part of the problem here is that the description of the sampling locations (and the plots on the map) is too vague to know the spatial extent and juxtaposition of the different habitats. But even still, given the nature of typical forest plantations, the spatial extent of the sampling is unlikely to cover the broad extent of the landscape.

The study takes a rather convoluted route to test apparent competition. It uses statistical models of "training" sample data to predict parasitism rates and tests those against observed rates based on validation sample data. I have 2 issues with this.

First, while the models are statistically significant, the predictions explain less than 25% of the observed rates. (Note: the description of the method used to develop predictions and observations is rather opaque, again plagued by writing that is jargon laden and not very clear.) The authors claim that this is a positive demonstration of apparent competition, but that becomes a matter of subjective judgment. From where I stand, 75% of the variation is not explained by the proposed dominant mechanism. So, reliable prediction has not been achieved, by my reckoning, and so the study suffers from rather than solves the same problems laid out in the first 2 paragraphs of the Introduction. The problem is that there are no a priori, objective criteria for falsification provided. I don't buy the tired old arguments that such is the nature of ecological data. No manager in the right mind would manipulate the environment based on such a high risk (75%) of lack of outcome.

Second, why not present data in a more direct and simple way? If apparent competition is operating, then one should expect that reducing lepidopteran populations in the insecticide treatment habitats should result in a rise in parasitism rate in the adjacent habitats. Does this in fact arise? I appreciate that one needs to account for variation in food web connections between prey and parasitoid species, so some weighting by species may need to be applied. But it seems to be that the presentation of results would provide a clearer test of the hypothesis. This is especially so because the analyses presented in the current version of the MS tends to obscure more than enlighten. That coupled with the opaque writing makes it difficult to judge the reliability of the data.

In the end, this will be a rather complex story, regardless. It is not easily told within the tight confines of Nature Communications. My sense is that the study would be better served by a more disciplinary venue that doesn't relegate important methodological details to on-line supplemental information.

Response to Referees

Reviewers' comments (bold text):

Reviewer #1 (Remarks to the Author):

A. Summary of the key results

The authors show that food web data can predict changes in the attack rates of natural enemies on hosts in neighbouring habitats, following natural and experimental changes in host abundances.

B. Originality and interest. The novelty of the manuscript comes from the experimental approach, the investigation of 'across-habitat' effects, and the fact that it is applied across a very large number of interacting species. Previous investigations (cited by the authors) have looked at across-boundary effects for small sets of sets of interacting species and/or have used observational rather than experimental methods. It seeks to address important ecological questions (Can we predict dynamics from 'static' food web patterns? Do shared enemies determine the relative abundance of species?) as well as applied questions (Can mobile, shared consumers link population dynamics between man-modified and 'natural' habitats?).

Author response: We are grateful to the reviewer for their support of the novelty of our work.

C. Data & methodology

This is a hugely impressive, large-scale dataset. The effort that has gone into compiling these data should not be underestimated! Methods of data manipulation and analysis are pretty complicated and difficult to follow - Extended Data Figure 2 was useful to summarise and explain the process and I wonder if this could be used in the main text?

Author response: We are grateful for the reviewer's appreciation of the dataset, and have now included this figure as the first figure in the main text.

D. Appropriate use of statistics and treatment of uncertainties

All the analysis is limited to parasitism rates but ultimately for apparent competition these should translate into changes in abundance. I understand the authors' logic (lines 88-94) for concentrating on parasitism rates, but why not also analyse and report effects on abundance? This would tell us whether the effects of apparent competition are actually having an impact on

abundances, in the context of the 'other drivers' of host population size that the authors mention; fundamentally this is how apparent competition is defined. Surely if you want to make the case that apparent competition is important in this system (the main claim of the manuscript) then you need to test if it actually influences host abundances? Otherwise we are left thinking that its effect, though statistically significant, may be minimal - yes it explains 15.6% of variation in parasitism rates but how does this modest contribution translate into population dynamic terms, relative to those other potentially important factors influencing host abundances?

Author response: We agree with the reviewer and thank them for this suggestion. We ran this analysis and found that in fact, expected parasitism rate predicts change in host abundance better than it predicts observed parasitism rate. This is likely because the time scale of our 'before' and 'after' collections was better suited to detecting the *effects* of parasitism (on abundance) rather than detecting the parasitism event itself. We have now changed the manuscript throughout to highlight this result, since the reviewer is correct that the importance of apparent competition as a structuring force is strengthened by demonstrating both the mechanism (i.e. effects on parasitism rates), and also the outcome (changes in host abundances) across communities. See Fig. 5g and Supplementary Table 9.

E. Conclusions: robustness, validity, reliability

Figure 3 is a key figure that seeks to encapsulate the main result but it is not hugely convincing! While I trust the authors' models and analysis skills, it would be good to have strong reassurance from plotting of data that the statistically significant patterns identified in the models are valid and are biologically as well as statistically significant. This is particularly the case when the models are complex, as here.

Figure 3 also needs better explanation! This may be my failing but I think the data points are individual host species (separated by site), according to the main text (not explained in the legend)- but why then are there so few data points? I also don't follow why the residuals vary between -2 and +2. Rates (both predicted or observed) must surely vary between 0 and 1 by definition, so how can you get a rate residual of -2?

Author response: The reviewer understood correctly that these are host species in sites. The reason that there are few points is that each point represents a species within a site for which individuals were collected and successfully reared at both the t and $t+1$ time steps, and for which at least one individual was parasitized in the first time step (these all being

necessary to calculate an expected parasitism rate for that species within a site; see Eq. 3). Despite enormous collection and rearing efforts, the numbers of species for which it was possible to calculate an expected parasitism rate at each site were far lower than the number of species collected at each site in at least one of the time steps. We expect this to be the typical scenario that will occur with all field experimental manipulations of species interaction networks under real-world conditions, due to turnover of species and interactions. However, we ended up with enough data points for robust analysis, and the expected parasitism rates for these species within sites are based on the potential for apparent competition with every other host species that occurred in the site, so the few data points do not prevent us from testing community-wide apparent competitive effects on future parasitism rates. To clarify this we have added (lines 263-271):

“The number of species for which it was possible to calculate an expected parasitism rate at each site was far lower than the number of species collected at each site. In Fig. 5, each point represents a species within a site that was both collected and successfully reared at both time steps, and was parasitized in the first time step (see Eq. 3, Fig. 1i). However, the expected parasitism rates for these species within sites are based on the potential for apparent competition with every other host species that we collected in the site, whether or not it was parasitized or reared successfully, so the few data points are predicted by data from the entire network, and allowed us to test community-wide apparent competitive effects on future parasitism rates.”

We have also included in the caption for this figure (now Fig. 5):

“Each point represents a species within a site that was collected and successfully reared at both time steps, and was parasitized in the first time step (see Eq. 3, Fig. 1i).”

The residuals in Fig. 5 a,b,c, and d do not range between zero and 1, despite being parasitism rates, because the data are not back-transformed after using the canonical logit-link transformation for the glmm with a binomial distribution. These linked values can take values from negative to positive infinity. We originally presented linked rather than back-transformed data because the point of these figures is to show how well the model explains variation in the data, and generalized linear model equations pertain to the linear predictor (η), which has already undergone a link function. However, we have now also added plots showing the raw data for observed and initial parasitism rates (Supplementary Fig. 2a,b),

which we added in response to a comment from Reviewer 3). We have also added the following explanation in the Figure 5 caption as to what the residuals represent and why we have plotted them:

“Residuals are deviations of the logit-linked data from the best model, and show the unexplained variation in the data remaining once variation due to the other fixed and random effects in the model have been accounted for. That is, they show the variation in the data that we are hoping to explain with the predictor on the x-axis.”

F. Suggested improvements

It would not be realistic or helpful to compile more data to improve the manuscript. The ms could be improved with better explanation and exposition of results as described elsewhere, and by analysing the implications for host abundance as well as parasitism.

Author response: We have re-written the introduction to give a better overview of the methodological approach, and have now included the methods figure as the first figure in the main text. We have also expanded the results section to add much greater detail and clarity about our findings in both the text and figures. We have included the suggested analysis of the implications for host abundance.

G. References

These are fine and give appropriate credit to previous work in this area.

H. Clarity and context: lucidity of abstract/summary, appropriateness of abstract, introduction and conclusions

The analytical approach taken is complex (through necessity, I think) and the reader has to work hard to understand the methods used - I really needed the Supplementary Material to get to grips with both the predicting of indirect interactions, and the modelling. I think the text could be improved and simplified to help with comprehension.

For example:

- the Abstract does not really include a clear description of what the authors have actually done/investigated e.g. make it clear that this study involves a large scale manipulative experiment so that the results stand on their own.

Author response: Our abstract is now shorter, due to the length limits for Nature Communications; however, we have now made it clear that the study involves a large scale manipulative experiment by adding:

“Here, using a large field experiment, we show that predicted apparent competitive effects of herbivore species on each other can significantly explain future parasitism rates and abundances of herbivores. These predictions were successful even across edges between natural and managed forests, following experimental reduction of herbivore densities by aerial spraying over 20ha.”

- Lines 95-96 - this is the key sentence to encapsulate the results but it is not easy to understand on its own. This is the first mention of a 'model' and at this point the reader doesn't really know what is being predicted from what.

Author response: We have now lengthened the introduction to give a complete overview of the methodological approach (see lines 110-168), and have also put the methods figure in the main text as Fig. 1. As part of the introduction, we have now added a sentence to inform the reader that the ‘models’ discussed in the results are statistical models (lines 164-167):

“Finally, we used statistical models to test whether these expected parasitism rates predicted the observed parasitism rates at the time step after spraying (Fig. 1j), as well as changes in abundance between time steps (Fig. 1k).”

Minor comment:

Line 324 Parasitism "rates" greater than one are in fact possible: superparasitism or multiparasitism, observed in many parasitoids. However their outcome won't be documented with data collected in the way the authors describe, based on successful emergence rather than dissection of hosts or sequencing of parasitoids within hosts.

Author response: We have changed that sentence (lines 579-581) to read:

“We reduced these expected parasitism rates to 1 in our analysis because our definition of parasitism rate was parasitized hosts / total hosts (i.e. host centric), so rates greater than one are not possible.”

Reviewer #2 (Remarks to the Author):

This manuscript is about apparent competition between Lepidoptera species due to shared parasitoids across a habitat boundary, and in response to changes in host abundance. The two habitat are plantation and natural forest. The change in host abundance is due to

seasonal variation, and a one-time insecticide spray. The authors found the host and parasitoid community to be the same in the two habitats. There was strong apparent competition. The change in attack rate that occurred with change in host abundance in one habitat (and also presumably parasitoid abundance) could be predicted based on current attack rate, demonstrating that apparent competition was an important force linking what happens in one habitat with what happens in the other.

Overall it is a very interesting study, both as basic ecology, and as it applied to our understanding of the impact of cultivation in the landscape, and conservation biological control. However, I found the rationale for the study and presentation of the results unclear and in some cases misleading or oversimplified.

Author response: We thank the reviewer for such thorough comments on the manuscript. We have now considerably expanded the introduction and results sections to clarify the methods, results, and rationale for the study.

More detailed comments:

I have made notes on the attached PDF, and present just the main points here:

Author response: We have carefully considered all the comments in the attached PDF, and have incorporated those that were still relevant after re-writing the introduction and results.

There was some information presented in the methods that should have been presented in the introduction and results. Such as a simple statement of hypotheses being tested, and what data were collected.

Author response: We now clearly state the hypotheses being tested in the introduction, lines 93-102:

“Here we test: 1) whether apparent competition determines community-wide parasitism rates and changes in herbivore abundance in host-parasitoid interaction networks (‘food webs’) at the interface between native and plantation forests. In so doing, we also test: 2) whether the future parasitism rate and abundance of each herbivore host species in the community can be predicted from quantitative food-web data on parasitoid overlap between hosts, as well as information about changes in abundance of all other hosts. We further test: 3) whether such predictions are possible across a habitat edge, or whether the edge hinders parasitoid movement or changes parasitoid host selection such that predicted

apparent competitive linkages between herbivore populations on either side of the edge are not realized.”

We now state as simply and clearly as possible what data were collected and our analytic approach in lines 110-168. We believe that though the introduction is now necessarily longer, the methods and rationale for the study are much clearer.

I strongly recommend explaining to us the connection between this ms and references 24 and 7. It is the same research system and collection style Is it the same data? It is confusing to try to understand the apparent competition part of the story without the known information about spillover and the match between the two communities. Also, it is hard to see what is presented here that isn't presented there (outside of the predictions of attack rate calculations which is for sure a main point of this ms).

Author response: Reference 24 (Peralta et al. 2014, Ecology) is an analysis of the quantitative food web data collected at the ‘training’ sites in this study (not the ‘validation’ sites). However, in the Peralta et al paper, there is no experimental herbivore reduction component, and the analysis and hypotheses tested are completely different.

In another paper (Peralta et al 2015, Journal of Animal Ecology, “Phylogenetic diversity and coevolutionary signals among trophic levels change across a habitat edge”), we also analyze the data from the training sites only. However, again the hypotheses tested in that paper (whether the matching of host and parasitoid phylogenies differs in native vs. plantation forests) are very different from those tested here, and there is no overlap in the analysis.

Reference 7 (Frost et al 2015, Ecology) was a study of parasitoid movement across the plantation-native forest edge at the sites that we call the ‘validation sites’ in the current ms. As such, it does not at all overlap with, but nicely complements the data described here, and we have taken the reviewer’s suggestion and added a more explicit reference to that study in the introduction (lines 102-109):

“We conducted a simultaneous study of adult parasitoid movement between the two forest habitats considered here²¹. That study showed that parasitoids of many of the same species considered here moved between habitats throughout the season. However, more individuals moved from plantation to native forest than in the other direction, likely due to the higher productivity of plantation relative to native forest²¹. Thus, it could be

that apparent competitive effects are asymmetrical between habitats, with stronger effects from herbivores in plantation forest on herbivores in native forest.”

In the Frost et al. paper we also describe the caterpillar collection methods described here. It was, in fact, the same collection as described in the current manuscript for the ‘validation sites’, but in that paper we do not present the vast majority of the data that we collected from our caterpillar collection and rearing efforts. We present those data here in the form of the quantitative food webs for each validation site (see Fig. 1g,h). These have not previously been published, and represent a huge effort in the field and laboratory. We describe the caterpillar collection methods in the Frost et al. paper only to explain how we were able to know which of the parasitoid species that we had trapped moving (as adults) across the edge actually parasitize Lepidoptera in the system, but in that paper we do not present any data on parasitism rates or any quantitative food web data. In the Frost et al paper we also present the effects of our herbivore reduction experiment on caterpillar abundances, but here, as in that paper, that result is presented to prove the method, and not as a novel and interesting result.

Intro paragraph:

I found this first paragraph (mostly lines 34-41) to be not very informative about the study. I made suggestions, but my overall suggestion is to rewrite the intro in more straight forward language.

Author response: We have taken the reviewer’s suggestion, and have almost entirely re-written the introduction to include less jargon, to explain the concept of apparent competition more clearly, and to explain much more clearly our methodological approach, and how our methods expand on what is already known in the field.

It would also be easier to understand the paragraph is paragraph might work better if the idea of flow between production and non-production was introduced earlier (now in line 50).

Author response: We have now changed to the Nature Communications format, which requires a shorter abstract, and has more room in the introduction, so we have deleted the paragraph that this comment refers to. However, we still introduce the concept of predator flow between production and non-production landscapes after the concept of apparent competition and the question of whether it is a community structuring force, since our test of cross-habitat apparent competition between managed and

natural forest depends on first testing whether or not apparent is an important community-level structuring mechanism.

Line 53: Land use changes also bring in new species.

Author response: This is true, but as it is not a central focus of our manuscript, and as the introduction is already on the long side, we have not included this point.

Lines 72-74: I see how the spray changes host abundance in one habitat and not the other, but the natural seasonal changes must be the same in both habitats. It isn't clear to me what the specific hypotheses are about apparent competition and changes in the host abundances

Author response: Apparent competition is a change in the abundance of one species resulting from the change in abundance of another species with which it shares a predator. Thus, in order to detect apparent competition if it actually does occur, we needed to make sure that host abundances changed between time steps. Normally insect herbivore abundances change at least slightly throughout a season, so our study would have been possible without our herbivore manipulation experiment, since yes, natural seasonal changes in host abundances should occur across all sites. However, our experimental herbivore reduction allowed a comparison of whether the efficacy of our expected parasitism rate as a predictor depended on the magnitude of change in host abundance. (For example, it could have been that expected parasitism rate was only an effective predictor for future parasitism rates when host abundances in the system changed drastically. We have now reworded our hypothesis statements (as mentioned above), so that we should cause the reader less confusion about experimental vs. natural changes in host abundance, since in fact that is not the most important question.

Line 77: If just edges were samples, how were the two habitats compared? (I figures out the answer to this later when I read the methods, but it should be presented here.

Author response: We now clarify early on that each 'site' was a habitat edge, and that sampling on either side of each edge allowed comparison of the two habitats (lines 114-116):

"Each site comprised samples from either side of a habitat edge between plantation Pinus radiata forest and native forest in New Zealand."

Lines 85-86: Wouldn't this also kill all the parasitoids in the Lep larvae? At least it would kill specific types (that were not adults at the

moment), which would change the food web.

Author response: The aerial spray would indeed have killed most parasitoids that were within hosts in herbivore reduction sites at the time of the spray. However, we attempted to time the spray so that it would roughly fall 'between' parasitoid generations, such that the adults from the first spring generation would have emerged, and would be mating and ovipositing at the time of the herbivore reduction treatment. See Supplementary Fig. 1, which shows that the herbivore reduction spray did indeed occur at the peak of adult parasitoid wasp activity at control sites, suggesting that we succeeded as far as possible to spray when most parasitoids present were in adult rather than larval form. Thus, we expect that generally the herbivore reduction treatment allowed us to test the effects of reducing mostly host abundances. However, the other two possibilities for what we may have accomplished are: 1) for some species the herbivore reduction treatment could have killed hosts with late instar parasitoids from the first generation; and 2) for other species the herbivore reduction treatment could have killed hosts with early instar parasitoids from the second generation. The latter (2) would create an effect on parasitism rates that is unpredictable based on shared parasitism (i.e. generate 'noise' in the results, rather than spuriously generate the results we observed). The former could generate a more extreme effect on parasitoid abundances (and therefore attack rates) than reducing unparasitized host abundances might have. However, it would still allow us to test the effect of reducing the abundance of shared parasitoids on attack rates on host species, proportionate to the amount that each host pair shares parasitoids. That is, it would still generate the desired effect of reducing the parasitoid subsidy from plantation forest to native forest, which would allow us to test whether the effects of such reductions on the host species in the native forest were predictable based on proportions of shared parasitoids.

Most importantly though, if the herbivore reduction treatment generated experimental artifacts, we would have observed different results in those sites compared with control (unsprayed) sites. However, the treatment factor was removed during model selection because it did not significantly alter the correlation between Expected and Observed parasitism rates. In other words, the patterns we observed following spraying were comparable with those when natural changes in abundance were used instead of spraying, whereby changes in the pool of shared parasitoids were an independent, dynamic response to altered abundance of hosts.

Lines 98-102: This seems strange to me because:

1) the experimental manipulation would kill leps in just the plantation (is that how it went)? and not the forest, where as the seasonal change in abundance would probably occur in both habitats.

Author response: See the comment above about seasonal vs. experimental change in host abundance. Essentially, it doesn't matter to our analysis that seasonal change in abundance would occur as well in the herbivore reduction sites, since the herbivore reduction would nonetheless cause more drastic changes in abundance than occurred through natural seasonal change. We now explain the herbivore reduction manipulation in the introduction, referring to the methods overview figure, and provide this rationale (lines 157-159):

“This allowed us to test whether our predictions of apparent competition performed equally for small, typical variation in host abundance (at control sites) as well as more dramatic changes, such as may occur during pest outbreaks or at plantation harvest.”

2) The spray would kill parasitoids that are in or on leps, so the parasitoid community would be different (also lines 209-211).

Author response: See our comment above on this point.

Lines 156-160: I think the general applicability of the results is exaggerated here because the match between the insects in an agricultural field and surrounding landscape is generally low, and parasitoids generally have narrow host ranges. Maybe it would work better when thinking of predators.

Author response: We already acknowledge in this section that “apparent competitive effects between habitats will be higher when the two habitats have high species similarity”, but the point of this paragraph is that prediction of the effects of apparent competition, regardless of the extent to which apparent competition occurs, should be equally successful whether or not predators move between habitats as frequently as in our system, since our index of the potential for apparent competition incorporates a prediction of which host species are likely to apparent competitively affect one another based on which parasitoids they normally share. So at an agricultural forest edge in which there was little parasitoid movement between habitats, our expected parasitism rate would still be an accurate predictor, even though it would largely incorporate within-habitat apparent competitive effects, rather than cross-habitat effects.

Lines 167-169: Not sure what you mean by regional vs site specific assessment of shared parasitism.

Author response: We mean that the d_{iAjB} values used in Equation 3 were calculated from different sites (the training sites) than the expected parasitism rates were calculated for (the validation sites). We hope that our changes to the introduction have clarified this distinction, so that this sentence in the discussion will be comprehensible.

Line 220: Why assume the parasitoids disperse a small distance? If they do, then doesn't that change the interpretation of the results quite a lot?

Author response: We respond to this comment below, where Reviewer 3 has asked a very similar question, in order to avoid repetition.

Line 551: Which one is habitat A?

Author response: Habitat A is the habitat where focal host i is found (see equations 2 and 3). Since we predicted parasitism rates for hosts in both plantation and native forest, habitat A took the value of “plantation” when the focal host was in plantation, and took the value of “native forest” when the focal host was in native forest.

Line 608: This figure is nice. Why hide it in the supplementary material?

Author response: We thank the reviewer, and now we have made it Figure 1 in the main text.

Reviewer #3 (Remarks to the Author):

This MS reports on an unprecedented landscape-scale manipulation to test a foundational idea—apparent competition—in community ecology. However, I have a lot of issues with the MS. First, and perhaps most importantly, the MS is not written in a way that is even remotely accessible to the broad readership of Nature Communications. It is laden with technical jargon and incomplete explanation of concepts and rationale.

Author response: We thank the reviewer for this comment, and we have re-written the abstract, introduction, and results in order to make them simpler, clearer, and less replete with ecology jargon.

The main prediction of apparent competition is that increases in the abundance of one species can indirectly decrease the abundance of another species through changes in species consumption by a shared natural enemy. Classic apparent competition assumes all species exist in the same habitat. This experiment tests this idea

using insect prey-parasitoid nature enemy system but adds an interesting twist: it looks at spillover effects of changes in prey species abundance in one habitat on prey species in and adjacent habitat. Mobile parasitoid species that shuttle between habitats are hypothesized to be the mediators. Indeed, the entire study is predicated on this key assumption of mobility. However, the study never reports data to validate this lynchpin assumption.

Author response: In fact, we have flight intercept trap data showing that 106 species of adult parasitoid wasps (which were the majority taxon of parasitoid individuals reared out of hosts), including most species that are included in this study, do indeed move across these forest edges. We cite the paper where we presented these data, but the reviewer is right that this is a key point to draw to the reader's attention, so we have now stated this previous result from the system more explicitly (lines 102-109):

"We conducted a simultaneous study of adult parasitoid movement between the two forest habitats considered here²¹. That study showed that parasitoids of many of the same species considered here moved between habitats throughout the season. However, more individuals moved from plantation to native forest than in the other direction, likely due to the higher productivity of plantation relative to native forest²¹. Thus, it could be that apparent competitive effects are asymmetrical between habitats, with stronger effects from herbivores in plantation forest on herbivores in native forest."

In addition, the analysis examines interactions merely within a very short distance from the edge between 2 habitats: it doesn't test whether there is spillover to the broader extent of each habitat. Therefore, the results could be a transient edge effect rather than an explanation of process across the landscape. Part of the problem here is that the description of the sampling locations (and the plots on the map) is too vague to know the spatial extent and juxtaposition of the different habitats. But even still, given the nature of typical forest plantations, the spatial extent of the sampling is unlikely to cover the broad extent of the landscape.

Author response: Dispersal distances for most parasitoid species are unknown, though field studies suggest distances ranging from tens of metres to tens of kilometres; Elzinga et al 2007, Basic and Applied Ecology). Therefore, our decision about what distance from the habitat edge at which to collect caterpillars was necessarily arbitrary. Given that parasitoid dispersal distance from a source (for our purposes a habitat

edge) is probably a long-tailed distribution, with many species moving at least a short distance across the edge, and fewer and fewer species and individuals arriving at further and further distances into the new habitat, we decided to select a distance from the edge just far enough that across all sites, the vegetation at that distance from the edge was distinct to that forest type. We state (lines 466-468), that “*we collected caterpillars at each site by establishing one 50 m transect in each forest type, 10 m from and parallel to the edge, which we designated as the last row of pine trees abutting the native forest.*” As we describe in the methods, each edge ‘site’ was at least 1 km from the nearest edge ‘site’. To clarify spatial extent of the forests we have added (lines 444-445)

“Each forest (plantation or native) was large enough to have an ‘interior’ location at least 400 m from all edges.”

It is true that we don’t test whether cross-habitat apparent competition occurs at any greater extent than this, though despite the short distance between transects in adjacent habitats (20 m), the vegetation composition was already very different in the two forest types (we refer to a detailed site description in a previous paper from the same system, lines 443-444). We certainly do not imagine that our results ‘cover the broad extent of the landscape’ other than that many landscapes are full of habitat edges, at each of which our results suggest that cross-edge apparent competition may be more or less important, depending on the degree to which parasitoids species both move across the edge, and recruit from hosts on either side of the edge. We have added the following sentence to the discussion to highlight that the effect we have observed is likely an edge effect (lines 399-405):

“Nevertheless, the cross-habitat apparent competitive effects that we found are likely an edge effect rather than a coupling of interior habitat populations. Cross-habitat apparent competition is mediated by mobile predators, and cross-habitat predator subsidies are likely to decrease with distance from the habitat edge towards the interior. That said, modern landscapes are often fine-scale mosaics of different land use types, with a proliferation of habitat edges that give prominence at a landscape scale to edge effects.”

The study takes a rather convoluted route to test apparent competition. It uses statistical models of "training" sample data to

predict parasitism rates and tests those against observed rates based on validation sample data. I have 2 issues with this.

First, while the models are statistically significant, the predictions explain less than 25% of the observed rates. (Note: the description of the method used to develop predictions and observations is rather opaque, again plagued by writing that is jargon laden and not very clear.) The authors claim that this is a positive demonstration of apparent competition, but that becomes a matter of subjective judgment. From where I stand, 75% of the variation is not explained by the proposed dominant mechanism. So, reliable prediction has not been achieved, by my reckoning, and so the study suffers from rather than solves the same problems laid out in the first 2 paragraphs of the Introduction. The problem is that there are no a priori, objective criteria for falsification provided. I don't buy the tired old arguments that such is the nature of ecological data. No manager in the right mind would manipulate the environment based on such a high risk (75%) of lack of outcome.

Author response: In response to the first reviewer's suggestion, we tested whether our expected parasitism rate could predict change in host abundance, and found that expected parasitism rate predicted 31% of the variation in change in host abundance, a result for which we present a plot of the raw data illustrating this correlation (Fig. 1g), which we tested with a simple linear model (Supplementary Table 9), since in this case the random factors did not explain any of the variation. We have also now included a plot of the raw data for the correlation between expected and observed parasitism rate (Supplementary Fig. 2a), in order to try to make it easier to judge the data. We have also tried to make our methods more transparent by putting the methods overview figure in the main text as Fig.1, and by re-writing the introduction to include a more detailed but straightforward overview of our data collection and analysis approach. We hope that this will make the methods much clearer.

Yes, we consider this to be a positive demonstration of apparent competition, since as an indirect effect, it would be very surprising if it explained anywhere near 100% of the variation in observed parasitism rates or change in host abundance. To explain the majority of variation in the abundance of an entire ecological community would essentially be to achieve a grand solution to all of ecology. The importance of this work for basic and applied ecology is that apparent competition has previously only been hypothesized to be an important community structuring mechanism, but this had never been tested at a community-level, and the two best tests so far (Morris et al 2004, *Nature*, and Tack et al 2011, *Journal of Animal Ecology*) have produced conflicting results as to the direction and

importance of this class of indirect interactions. Moreover, we show a statistically significant effect of apparent competition across different habitats, also for the first time. We believe it is quite remarkable across the noise of varying species' phenologies, not to mention environmental effects and other species interaction effects, that we still found that 31% of the variation in host abundances and 15% of the variation in parasitism rates could be explained by one single predictor variable - apparent competition. In terms of applied predictions, this is better than no knowledge of how important apparent competition might be at the community level, and is a foundation upon which other studies can build.

To address the comment that we provide no objective criteria for falsification, we decided to additionally test whether simply knowing initial parasitism rate was sufficient to predict observed parasitism rates and changes in host abundance. Temporal autocorrelation might mean that parasitism rates at an initial time step are a fairly good predictor of final parasitism rates and abundances, and calculations of the potential for apparent competition between species might not improve predictions much beyond this. We present the results of these analyses (Fig. 5b,f; Supplementary Tables 5, 10), which suggest that initial parasitism rate predicts neither observed parasitism rate nor change in host abundance as well as expected parasitism rate does. Therefore, the objective criterion we use for falsification is a statistical hypothesis test, which demonstrates within the accepted margin of error that apparent competition provides better predictive power than an equivalent model without apparent competition.

Second, why not present data in a more direct and simple way? If apparent competition is operating, then one should expect that reducing lepidopteran populations in the insecticide treatment habitats should result in a rise in parasitism rate in the adjacent habitats. Does this in fact arise? I appreciate that one needs to account for variation in food web connections between prey and parasitoid species, so some weighting by species may need to be applied. But it seems to be that the presentation of results would provide a clearer test of the hypothesis. This is especially so because the analyses presented in the current version of the MS tends to obscure more than enlighten. That coupled with the opaque writing makes it difficult to judge the reliability of the data.

Author response: We have taken the reviewer's suggestion, and now present an analysis of average parasitism rates by habitat, treatment, and timestep (Fig. 4b) as well as mean caterpillar numbers by habitat, treatment and timestep (Fig. 4a), which together give a good overview of the overall effects of the herbivore reduction manipulation. The reviewer is

correct that one might expect a rise in attack rates in native forest immediately following the spray, though this would not be an apparent competitive effect. Rather, it would amount to apparent mutualism if the hosts in plantations had been attracting parasitoids away from those in the native forest. Our added results show that this did not occur though. There was no significant effect of herbivore reduction treatment in the plantation on parasitism rates in the adjacent native forest. However, this test is so imprecise that a negative result does not necessarily mean that apparent competition (or mutualism) is not operating. Rather, it could mean that many hosts do not share parasitoids across the edge, or that they share parasitoids with host species for which the herbivore reduction experiment less drastically altered their abundances than for others (for example because of a 'sheltered' larval habit, e.g. leafrollers). Such effects would create noise, which could easily make even strong apparent competitive linkages undetectable. This is precisely why we used the food web information to predict (successfully) how large the apparent competitive impact on each species should be, given their sharing of parasitoids. As described above, we have now tried to make the writing and presentation of these results much clearer (see lines 236-299 and lines 344-353).

In the end, this will be a rather complex story, regardless. It is not easily told within the tight confines of Nature Communications. My sense is that the study would be better served by a more disciplinary venue that doesn't relegate important methodological details to on-line supplemental information.

Author response: The 'confines' of Nature Communications are a lot less tight than for Nature, which the ms was originally formatted for. Now that we have expanded the introduction, results, and display items to increase detail and clarity, we feel that the story can be well told in this format.

Reviewers' Comments:

Reviewer #2 (Remarks to the Author)

The authors made a very thorough revision of the manuscript based on all three reviewer comments. It is very much improved, and a pleasure to read.

Most of my concerns were related to how material was presented, especially the introduction which previously did not make the hypotheses clear or the relation of this study to related studies. The introduction was entirely rewritten so it is much more clear now, as is the general text throughout the ms.

I was also concerned with the timing of the spray for the experimental part of the study. The authors pretty much convinced me that the spray happens when most wasps are adults which would make it not much of a problem. And given the results I guess that it could not have been a big problem.

I was also concerned with the mobility of the parasitoids and the assumption that they move short distances, given the spatial scale of the experiment. The authors added a reference that gives a large range of possible movement rates (which is likely true), but imply that since some parasitoids move short distances (less than 10s of meters), the ones in the study should. Given the results it must be that many do indeed only move at that scale, but the assumption still seems wrong and likely many parasitoids are also moving much more (and some less).

Finally- there is the generality of the result. The main point that apparent competition can be tested and detected at a community scale is strong. It also is clear that in this case it transfers between habitat types, which is an exciting result. However, as I mentioned in my original review, only some types of habitat pairs share parasitoid species to the extent found in this study. The Authors mention this briefly in the discussion but it would be easy to miss.

A few smaller comments

-In the introduction the new paragraph on the types of apparent competition, and especially apparent mutualism can be shorter because this study doesn't really get at that detail later in the ms.

-Also related to that paragraph, on lines 74-75- I don't think the lack of studies showing apparent mutualism means that it is less common in nature. I suggest leaving that part of the sentence out.

Line 93 says that the study tests whether apparent competition "determines" community wide parasitism rate. Determines is too strong a word because parasitism rate must also be due to other things as well.

Reviewer #3 (Remarks to the Author)

This MS is much improved over the last version. The analyses to test for apparent competition make sense now. I am satisfied that the authors have adequately addressed my concerns.

One small quibble. The statement made in the first sentence of the Introduction is plain wrong, as written. Ecologists have many methods to predict. The problem is that the methods have not been quantitatively evaluated in a wide number of settings.

There is, however, no need to get into all this. I would dispense with the first sentence altogether and start with the second sentence. It makes for a more logical lead-in.

Response to Reviewers

REVIEWERS' COMMENTS:

Reviewer #2 (Remarks to the Author):

- 1) The authors made a very thorough revision of the manuscript based on all three reviewer comments. It is very much improved, and a pleasure to read. Most of my concerns were related to how material was presented, especially the introduction which previously did not make the hypotheses clear or the relation of this study to related studies. The introduction was entirely rewritten so it is much more clear now, as is the general text throughout the ms.**

Author response: We thank the reviewer for these very positive comments.

- 2) I was also concerned with with the timing of the spray for the experimental part of the study. The authors pretty much convinced me that the spray happens when most wasps are adults which would make it not much of a problem. And given the results I guess that it could not have been a big problem.**

Author response: We agree that the results are the strongest evidence that this was not a problem. Specifically, the fact that the herbivore reduction treatment factor was removed during model selection because it did not significantly alter the correlation between Expected and Observed parasitism rates suggests that the spray treatment did not generate unexpected effects on parasitoid populations that worsened the model fit. In other words, the patterns we observed following spraying were comparable with those when natural changes in abundance were used instead of spraying, whereby changes in the pool of shared parasitoids were an independent, dynamic response to altered abundance of hosts.

- 3) I was also concerned with the mobility of the parasitoids and the assumption that they move short distances, given the spatial scale of the experiment. The authors added a reference that gives a large range of possible movement rates (which is likely true), but imply that since some parasitoids move short distances (less than 10s of meters), the ones in the study should. Given the results it must be that many do indeed only move move at that scale, but the assumption still seems wrong and likely many parasitoids are also moving much more (and some less).**

Author response: We agree that again, the results of the study are the

strongest argument that the scale of the experiment was appropriate; that is, that cross-edge predictions at this scale were successful. This is not to say that this is necessarily the only spatial scale over which we would have been able to detect apparent competitive effects, since the reviewer is probably right that many parasitoid individuals would move farther than 10 metres across the edge, and some (probably many) would move less. Indeed, it would be interesting to know how far from the edge cross-edge apparent competitive effects are detectable. However, we wonder what the reviewer's basis is for saying that our finding that many parasitoids moved only short distances across the edge "seems wrong". Metacommunity models often assume that parasitoid dispersal between patches is diffusive, such that more individuals arrive the closer a patch is to the 'source' (e.g. Comins et al. 1992, *Journal of Animal Ecology*). Field studies have further corroborated this assumption. For example, Kruess and Tscharntke (2000, *Oecologia*) found that parasitoid species loss with habitat fragmentation greatly increased with increasing isolation of remaining fragments, with the steepest decrease in parasitoid species occurring within the first approximately 10 metres of isolation distance (see Fig. 1D in that paper). (Also see other empirical papers similarly suggesting very short parasitoid dispersal distances, referenced in Elzinga et al. 2007; *Basic and Applied Ecology*.) Thus, we assume that parasitoid dispersal distance from a source (for our purposes a habitat edge) is probably a long-tailed distribution, with many species and individuals moving at least a short distance across the edge, and fewer and fewer species and individuals arriving at further and further distances into the new habitat.

4) Finally- there is the generality of the results. The main point that apparent competition can be tested and detected at a community scale is strong. It also is clear that in this case it transfers between habitat types, which is an exciting result. However, as I mentioned in my original review, only some types of habitat pairs share parasitoid species to the extent found in this study. The Authors mention this briefly in the discussion but it would be easy to miss.

Author response: We currently mention three times in the discussion that cross-habitat apparent competitive effects are more likely when habitats are similar and large amounts of species are shared:

-line 421: "These impacts would be particularly likely when the edge between habitats has a low structural contrast (such as between our two forest types) that does not impede consumer movement³³, and when species overlap among habitats is high, as was the case in this system"

-line 426: “indirect effects on native communities are likely to be greatest when more natural enemy species are shared with productive systems (i.e. when the habitats have high species overlap)”

-line 455: “Although apparent competitive effects between habitats will be higher when the two habitats have high species similarity, predictions should work even when there are fewer shared species across the edge, such as at an agroecosystem-forest edge, because the level of regional parasitoid overlap is incorporated into the expected parasitism rate”

We have also added a fourth mention of this in response to an editorial request to clarify the subject of a sentence:

-line 446: “This stronger cross-habitat apparent competition between hosts when the habitats and their species are more similar ...”

We therefore hope that the reader will not miss this point, and are reluctant to state it again a fifth time at risk of sounding repetitive.

A few smaller comments

- 5) In the introduction the new paragraph on the types of apparent competition, and especially apparent mutualism can be shorter because this study doesn't really get at that detail later in the ms.**

Author response: We are concerned about shortening this, since it is already a very short summary of a large body of literature, and although we don't test by which mechanism apparent competition is occurring in our study, we think that this paragraph provides the reader important background theory with which to interpret our results. Given that our manuscript is not over the length limit, we would like to keep this paragraph as it is, if the Editor agrees.

- 6) Also related to that paragraph, on lines 74-75- I don't think the lack of studies showing apparent mutualism means that it is less common in nature. I suggest leaving that part of the sentence out.**

Author response: We agree and have now deleted “suggesting that it may be less common in nature” from that sentence.

- 7) Line 93 says that the study tests whether apparent competition "determines" community wide parasitism rate. Determines is too strong a word because parasitism rate must also be due to**

other things as well.

Author response: We have now changed “determines” to “influences”.

Reviewer #3 (Remarks to the Author):

- 1) This MS is much improved over the last version. The analyses to test for apparent competition make sense now. I am satisfied that the authors have adequately addressed my concerns.**

Author response: We thank the reviewer for these positive comments.

- 2) One small quibble. The statement made in the first sentence of the Introduction is plain wrong, as written. Ecologists have many methods to predict. The problem is that the methods have not been quantitatively evaluated in a wide number of settings. There is, however, not need to get into all this. I would dispense with the first sentence altogether and start with the second sentence. It makes for a more logical lead-in.**

Author response: We have now deleted this sentence, and we start with the second sentence.